# Non-Destructive Techniques for the Condition and Structural Health Monitoring of Wind Turbines: A Literature Review of the Last 20 Years

**DOI:** 10.3390/s22041627

**Published:** 2022-02-18

**Authors:** Marco Civera, Cecilia Surace

**Affiliations:** Department of Structural, Geotechnical and Building Engineering (DISEG), Politecnico di Torino, Corso Duca degli Abruzzi 24, 10129 Turin, Italy; cecilia.surace@polito.it

**Keywords:** structural health monitoring, condition monitoring, damage detection, fault diagnostics, non-destructive testing, artificial intelligence, wind turbine, wind farm, blade monitoring

## Abstract

A complete surveillance strategy for wind turbines requires both the condition monitoring (CM) of their mechanical components and the structural health monitoring (SHM) of their load-bearing structural elements (foundations, tower, and blades). Therefore, it spans both the civil and mechanical engineering fields. Several traditional and advanced non-destructive techniques (NDTs) have been proposed for both areas of application throughout the last years. These include visual inspection (VI), acoustic emissions (AEs), ultrasonic testing (UT), infrared thermography (IRT), radiographic testing (RT), electromagnetic testing (ET), oil monitoring, and many other methods. These NDTs can be performed by human personnel, robots, or unmanned aerial vehicles (UAVs); they can also be applied both for isolated wind turbines or systematically for whole onshore or offshore wind farms. These non-destructive approaches have been extensively reviewed here; more than 300 scientific articles, technical reports, and other documents are included in this review, encompassing all the main aspects of these survey strategies. Particular attention was dedicated to the latest developments in the last two decades (2000–2021). Highly influential research works, which received major attention from the scientific community, are highlighted and commented upon. Furthermore, for each strategy, a selection of relevant applications is reported by way of example, including newer and less developed strategies as well.

## 1. Introduction

There is a general consensus from technicians, political leaders, and public opinion alike that the worldwide energy sector should shift to more sustainable sources. 

Wind power is widely considered one of the best options in this sense. As for any energy source, it has its own advantages and limitations; for instance, it is an intermittent source, not dispatchable on demand but rather subject to the fluctuating nature of meteorological conditions. Nevertheless, it is fully renewable and highly sustainable, with minimal environmental impact when compared to traditional fuel power. However, wind turbines (WTs) come with both worker and public safety concerns. 

In case of mechanical faults, turbine nacelle fires may erupt. Due to their height, these can be dangerous to extinguish, while releasing toxic flumes and potentially causing secondary fires in their immediate surroundings.

The risks are even more evident for structural collapses. These can be due to global or local failure mechanisms. The first case can be caused by a failure at any point along the tower height or its complete toppling due to foundation issues. In this instance, not only the wind turbine but also nearby structures can be damaged in the collision.

Even in the case of local failures, the consequences may be particularly severe, especially for turbines located near highly-populated areas. These failures include the detachment of the rotor from the nacelle or of the whole rotor–nacelle ensemble from the tower. However, the most well-known structural risk concerns blade brakes.

Indeed, according to Ref. [1], blade failure is the most common risk, accounting for ~65% of all incidents (when the detachment of both the full blade or part thereof are accounted for together). Even the loss of some smaller blade components may endanger people’s safety, due to their potential high impact velocity and the long distances they can cover when carried by strong winds. For this reason, WT blade monitoring will be particularly detailed in this discussion.

Thus, wind turbines require constant integrity and safety monitoring. This can be achieved with automated approaches, based on Artificial Intelligence (AI). However, these AI-based diagnostics strategies require damage-sensitive features for data-driven anomaly detection. These should ideally be retrieved from the operating WT in a non-destructive, non-invasive fashion.

To this aim, this paper reports a broad overview of non-destructive evaluation (NDE) approaches and the respective techniques (NDTs) for structural health monitoring (SHM) and condition monitoring (CM). Due to the very large quantity of published documents on this subject, vibration- and SCADA-based approaches will not be included here; instead, they will be covered in a dedicated review in the near future. 

The remainder of this paper is structured as follows. Section 2 describes the historical and current context of the WT and wind industry. Section 3 recalls the main components of any generic wind turbine and discusses their implicit risks and the main causes of damage/failure. Section 4 briefly describes the main sub-fields of applications for both CM and SHM in WTs. Section 5 lists the main NDE strategies for the structural and mechanical components. Section 6 reports some final discussions and suggestions. Finally, the conclusions end this paper.

## 2. Context: The Worldwide Politics and Economics of Wind Turbines

### 2.1. Climate Change and the Political Stance on Sustainable Energy Sources

The need for sustainable and renewable energy sources (also known as green energy) originated mainly in the last decades. This was due to the raising concerns about the mid- and long-term effects of human activities on the environment, on a global scale.

For this reason, in 1980 the World Meteorological Organization (WMO) organized the first world conference on climate “to prevent potential man-made changes in climate that might be adverse to the well-being of humanity” [2].

In the following years and up to the present day, numerous UN climate conferences and meetings have been held. Among these, the 1997 UNFCCC Climate Change Conferences resulted in the famous Kyoto Protocol, where binding milestones were set for the reduction in harmful emissions by industrialized countries. In 2015, these meetings led to the Paris Agreement, where 196 countries agreed to the goal of limiting the increase in the global temperature to less than two Celsius degrees above pre-industrial levels. The most recent UN conference on climate, named COP26, was ongoing at the time of writing. Therefore, there is pressure from both public opinion and decision-makers to subsidize green energy producers to extend their market share.

In this regard, the International Renewable Energy Agency (IRENA) recently published in its latest annual report [3] the statistics on the global renewable energy generation capacity. According to the data collected, the global renewable generation capacity amounted to 2799 GW at the end of 2020. In the same year, wind energy took second place with a capacity of 733 GW (26%), preceded only by hydropower sources (1332 GW or 48%) and ahead of solar energy (714 GW or 25%, including both photovoltaic and concentrated solar power). In more detail, onshore WTs and wind farms produced 699 GW, almost on par with the total solar energy production by itself, while offshore sites added a further ~34 GW. 

Furthermore, solar and wind energy continued to dominate the expansion of renewable capacity. Out of a total increase in the total renewable generation capacity by 261 GW (+10.3%) in 2020, solar energy continued to drive capacity expansion, with an increase of 127 GW (+22%), yet very closely followed by wind energy with 111 GW (+18%). Indeed, in most industrialized countries, due to the limited possibility of new large hydroelectric installations, the increased demand for renewable energy is largely covered by these two sources, with wind energy production exceeding its solar counterpart in many European countries, e.g., Italy, according to the most recent data [4]. From a historical perspective in the European Union, the wind power generation capacity has been well above solar photovoltaic capacity since the early 2000s, overtaking the fuel oil capacity in 2007, nuclear energy in 2013, hydroelectric in 2015, and coal in 2016, remaining only behind natural gas as of 2017 [5]. 

### 2.2. The Current and Near-Future Economic Impact of Wind Power

At the world scale, according to the latest data from BloombergNEF, wind power developers around the world commissioned a record 96.7 GW of installations in 2020, up 59% from 60.7 GW installed in 2019. This increase in capacity was mainly due to the surge in installations in China and the United States [6]. Specifically, 2020 set a record year of wind growth for China, with a 36% year-on-year increase in WT installations [7]. In the same year, the USA installed a record 14.2 GW, more than in any other year so far [8]. Other non-European high-GDP countries strongly committed to the transition towards wind energy include Japan (where the Japan Wind Power Association declared 2020 as the best year for capacity addition in the country’s history [9]), Australia (where wind energy supplied 35.9% of the clean energy in 2020, remaining the leading renewable source and setting a record-breaking year [10]), India (with a year-on-year increase of +5.9%, reaching a total of 37.7 GW installed nationwide [11]), Brazil (where the installed capacity increased from 1 to 18 GW from 2010 to 2020 [12]), and South Korea (where the government Green New Deal, announced in July 2020, set a goal of 12 GW of wind capacity by 2030, also announcing the largest offshore wind farm in the world, to be built in the South Jeolla province [13]). 

At the European scale, the 5-year outlook reported in Ref. [14] considered a growth of 15 GW p.a. as a “realistic expectation”. It should be stated, furthermore, that new installations (as of 2020) of offshore WTs strongly exceeded onshore ones in many North Sea countries (1493 MW vs. 486 MW in the Netherlands, 706 MW vs. 152 MW in Belgium, and 483 MW vs. 115 MW in the UK [14]); the aggregate onshore and offshore installations are portrayed in Figure 1. This makes economic sense due to the more stable and steady wind flows in the open sea and the lower acoustic and visual impact in comparison to their onshore counterparts. However, with the operation and maintenance (O&M) costs much larger for offshore production facilities, structural health and condition monitoring will be even more important.

In Mediterranean and southern European countries, this can act as a driving economic force for less-developed areas. For instance, Italy was the fifth country in Europe in terms of installed wind capacity, with more than 10 thousand MW of plants installed as of 2019 (mostly onshore), for a total of about 670 WTs [15]. Most of the wind farms (over 90%) are concentrated in the South and the islands of Sicily and Sardinia, due to the greater availability in these regions of adequately windy sites. At the moment, not even 1 MW of offshore installations are operative. However, the installation of the first offshore wind farm in Italy (in the Sicilian channel) is expected in the coming years.

The prospects in the medium to long term are very positive, especially due to the unique opportunity given by the Recovery Plan for Europe. According to the National Integrated Plan for Energy and Climate (Piano Nazionale Integrato per l’Energia e il Clima, PNIEC [16])], the installed wind energy capacity in Italy should reach approximately 19,300 MW by 2030, of which approximately 900 MW will come from offshore wind. This would guarantee an annual production of electricity equal to 40 TWh, which is 10% of the national gross electricity consumption.

### 2.3. Expected Returns and Benefits from WT Monitoring

For all the reasons discussed above, the impact of CM and SHM cannot be neglected from the perspective of more reliable, more cost-effective wind power production in the coming decades. From a risk management point of view, the NDE strategies and their related NDTs are required to reduce the number of both minor incidents and (fatal or non-fatal) severe accidents. From the economic perspective, the same approaches are intended to improve, rationalize, and automate as much as possible the maintenance routine. In general terms, the end goal is a paradigm shift from time-scheduled maintenance to condition-based maintenance, which is more flexible and avoids unnecessary deployments on-site. This can save man-hours and transportation costs. Fewer human operators on-site would also mean less exposure to dangerous conditions (especially for offshore wind farms). All these beneficial consequences would lower the O&M costs and therefore the total energy price, thus making the renewable energy from wind farms more cost-competitive in comparison to other traditional alternatives, such as fossil fuels. In this sense, the cost-efficiency of WTs is commonly evaluated in terms of (yearly or life-long) levelized cost of energy (LCOE), i.e., the total cost (per year or over lifetime) divided by the (annual or lifetime) energy production [17,18]. Decreasing O&M costs automatically increases the LCOE of new and already-existing installations [19].

In particular, for mechanical faults, the mean downtime can vary in a range from 6 to 15 days onshore. The cost of a gearbox bearing exchange ranges from EUR 15000 for a simple up-tower replacement to more than EUR 1 million for the substitution of larger (5 MW) gearboxes [20]. These figures are indicative of onshore installations; as mentioned earlier, failures in offshore WTs would result in even longer interruptions and costlier reparations [21]. McMillian & Ault [22] provided a detailed quantitative analysis of the impact of condition monitoring on O&M costs. In this regard, Figure 2 reports some indicative estimates.

Regarding (partial or total) structural collapse, while likely much rarer, this occurrence would result in the complete loss of the whole asset, plus the collateral damage; therefore, an SHM apparatus should be always considered jointly to the more common CM systems.

For all these applications, the expected gains of implementing an SHM/CM strategy can be evaluated in terms of its value of information (VoI [23]). Some examples can be found in Ref. [24] for the SHM of WT blades and in Refs. [25,26] for the CM of WT gearboxes and generators (in the same order).

**Figure 2 sensors-22-01627-f002:**
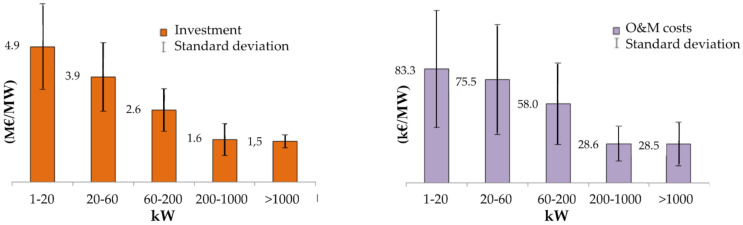
Bar chart of the estimated investment (**left**) and O&M (**right**) costs per MW of an onshore HAWT, according to the class of mean electric power produced. Based on data retrieved from Ref. [27].

## 3. Wind Turbines: Structural and Mechanical Components

There are several different typologies of “wind turbines”, the most common type being the so-called Horizontal Axis Wind Turbine (HAWT) systems. In this configuration, the rotation axis of the rotor is parallel to the ground. Specific attention must be paid to the orientation with respect to the wind direction, which is different from other types of wind turbines such as those with a vertical axis, whose orientation is independent of the prevailing wind direction. The upwind configuration is the most common one, with the rotor facing the incoming wind. 

In this review, only HAWT will be considered; the term “wind turbine” will therefore be used exclusively for this specific device hereinafter. In more detail, the only structural configuration of interest will be the classic towered HAWT, with one or more blades (usually three). Both on- and offshore turbines are included. Except for small wind turbines for private on- or off-grid energy production, any tower size and blade length are considered.

From an engineering perspective, any wind turbine is made up of:(i)static, load-bearing components;(ii)moving/rotating parts, needed to harness the wind’s kinetic energy and turn it into electricity.

The elements in (i) are generally referred to as the support structure. The components of (ii) can be further divided between slowly rotating elements (blades) and high-speed rotating mechanisms. These latter ones are all included in the rotor–nacelle assembly on top of this support structure.

These distinctions are essential since the blades and the support structure are fields of application for structural health monitoring, while condition monitoring, according to the common definition of the term, deals with machinery and rapidly moving components such as gears and bearings in the gearbox and generator.

Indeed, elements in both (i) and (ii) are subject to naturally occurring use and consumption and, therefore, can develop structural damage in the long run. Damage in the external structure will cause (partial and localized or global) collapse, while damage in the internal mechanisms will cause faults, disruption in the energy production, and potentially fire, explosions, oil leakage, or other events. 

Hereinafter, the main structural components and pieces of machinery of a towered, multi-bladed HAWT are reported and briefly recalled. A more detailed description can be found in the textbook of Hau & Von Renouard [28].

### 3.1. Components under Structural Health Monitoring

#### 3.1.1. The Tower

The tower is the main component of the support structure. Apart from foundation costs (which, as it will be discussed later, vary noticeably for on- and offshore structures), its cost can be up to one-fourth of the total (Figure 3).

The main parameter of the tower is its height. This is typically about 1.5 times the rotor diameter; generally, it is never lower than 20 m and can reach up to 150 m or more (for 10–12 MW outputs). In absolute terms, the higher the tower, the better the wind conditions in terms of intensity and constancy. The tower can be lattice or tubular. This second design choice has been more common since the mid-1980s. In this case, the tower is made of thin-walled steel conical parts of varying diameters and diameter-to-wall thickness ratios. These offer a practical and safer way for the survey teams to access the nacelle. Moreover, in comparison to lattice structures, there are fewer bolted joints to inspect and maintain. The tower diameter (maximum at its basis and minimum at its top) increases with the tower height; e.g., a typical 50 m-tall HAWT will have a diameter ranging from 3.5 m to 0.4 m [29]. 

From a vibrational and SHM perspective, the stiffness of the tower is the main parameter to be taken into consideration in evaluating the global dynamics of WTs due to the possibility of coupled vibrations between the tower and the rotor.

**Figure 3 sensors-22-01627-f003:**
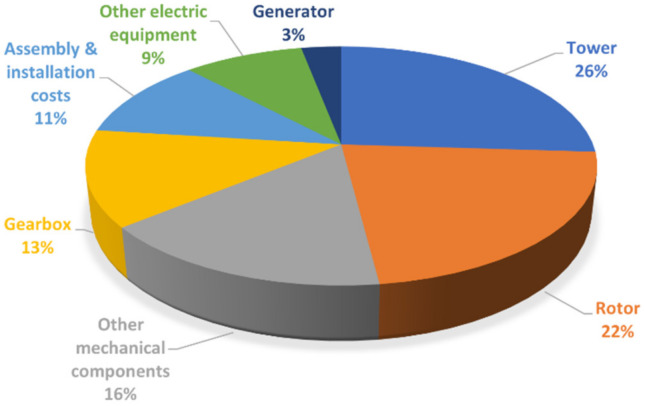
Estimated costs of a HAWT, as a percentage of the total and excluding foundations. Based on data retrieved from Ref. [30].

#### 3.1.2. The Substructure

It must be mentioned that offshore HAWTs, differently from their onshore counterparts, include a further group of structural elements, which are included below the platform and above the sea floor (Figure 4). These components are particularly at risk due to their location under water or—even worse—in the splash zone, immediately above/below the mean water level and highly subject to corrosion. Furthermore, being submerged, they cannot easily be visually inspected if not using divers or manned/unmanned underwater inspection robots. They are also subject to marine growth and other potentially damaging environmental conditions such as wave, tidal, and current forces.

#### 3.1.3. The Foundations

On- and offshore foundations differ sensibly. However, in both cases, the choice of the specific structural design depends on the location and site conditions. For example, the quality and strength of the soil are the main determinants affecting the size and shape of onshore foundations, while the depth of the water and the distance from the coast are the key factors for offshore turbines.

For onshore installations, both surface (shallow) and deep foundations are frequently used. In the wind farms dating back to the 1990s, square foundations with a constant thickness were commonly utilized. This solution, however, can lead to the formation of localized damage. Hexagonal and octagonal shapes subsequently became more common, even with variable thicknesses. The most modern designs use circular shapes, which allow the reinforcing bars to be positioned more homogeneously. This ensures a better dispersion of the forces induced by the soil–structure (or soil–pile–structure) interactions.

The foundation design, construction, and monitoring in offshore wind farms are more complex and challenging. The costs are much higher as well, absorbing a large percentage of the total expense of an offshore wind turbine. Specifically, Ref. [31] mentioned the offshore foundation costs to be 35% of the total project expenses. Refs. [31,32] estimated a cost from 352 EUR/kW (~19.6% of the total construction cost) for a water depth between 10 and 20 m, up to 900 EUR/kW (~35.8%) for transitional waters (40 to 50 m deep). For smaller HAWT closer to the coast (1–2 MW, with A water depth <30 m), monopile foundations are often encountered. Jacket/tripod substructures are more frequent between 25 and 50 m and for larger structures (2–5 MW) [33]. For deep waters (50–120 m and beyond), floating structures, anchored to the sea floor, are preferable. Conversely, gravity-based foundations (to the right side in Figure 4) can be found for offshore installations in very shallow waters but they are not widely used [31].

#### 3.1.4. The Rotor

The rotor is a crucial component of the wind turbine. Indeed, it is the most expensive mechanical component, up to circa one-fifth of the total cost [34]. Its design is one of the most critical and delicate phases, especially in terms of expected performances. Therefore, the economic feasibility of the whole system depends on these aspects.

The rotor consists of the blades and the hub from which they branch off. Generally, the rotor can be single-, double-, or three-bladed. The most common turbines include three blades arranged at 120° from each other; this is conventionally considered the optimal design.

The length of the blades determines the capability to convert high wind speed to low rotational speed and finally to electrical energy. As mentioned before, the diameter of the rotor governs the tower height and thus the overall size of the structure, including the foundations. Indeed, a larger rotor will produce a consequent increase in the energy produced but, on the other hand, will require a wider tower cross-section and more massive and/or deep foundations, thus increasing the construction costs.

For monitoring purposes, a “smart rotor” system should include several sensors (accelerometers, strain gauges, pitot tubes, pressure tabs, etc.), embedded and distributed along the blades [35]. Indeed, due to the prominence of the rotor blades, these components deserve to be discussed on their own.

#### 3.1.5. The Blades

WT blades are generally made of glass or carbon fibre reinforced polymeric (GFRP and CFRP) materials. E-glass fibres are particularly used as the main reinforcement in the composite material. 

The blade design is actually quite complex, with several different components and materials as pictorially described in Figure 5. This complexity makes them particularly susceptible to manufacturing defects, which indeed were estimated in Ref. [36] to account for ~51% of all blade damages (with debonding and voids in skin core being the most common defects at 20% and 18%, respectively). 

From a geometric perspective, the cross-section profile varies from root to tip, often also rotating around its main axis; all these aerodynamical aspects of their airfoil design characterize the blade lift-to-drag ratio and, by consequence, the wind-to-rotor efficiency. The internal reinforcements may include different sorts of load-carrying structural elements, such as shear webs, closed shells, box spars, or other geometries. This design choice affects the space available for the maintenance workers to operate.

As briefly mentioned earlier, the structural integrity of WT blades is of the foremost importance, due to the potential impacts of fully or partially detached blades with neighbouring structures.

### 3.2. Components under Condition Monitoring

#### 3.2.1. Drive Train (and Other Components Inside the Nacelle)

Positioned on the top of the tower, the nacelle is a cover housing, intended to shelter all the mechanical and electrical components installed inside from the external environment. These mechanisms include (for a conventional HAWT) the gearbox, rotor shaft, brake, and generator, all assembled together (Figure 6, adapted from Ref. [37]). These pieces are necessary for energy conversion and represent roughly between one-third and half of the HAWT total cost on their own [34]. For these reasons, they will be discussed in more detail separately.

Other mechanisms included in the tower and the nacelle are, for instance, the rotor yaw system and the control and power electronics systems.

The yaw system is responsible for the orientation of the nacelle–rotor assembly towards the wind, rotating 360° around the vertical axis. The control and power electronics systems are tools of fundamental importance to control the operation of the machine, manage the supply of electricity, and stop the system beyond certain wind speeds for safety reasons due to the excessive heat (generated by the friction of the rotor on the axis) and/or mechanical stresses. Therefore, they maximize the life of the system by ensuring the limitation of the fatigue of all components (fatigue that can result due to changes in wind speed and direction, the presence of turbulence, shutdown, and start-up cycles of the turbine, etc.). These systems include speed, position, temperature, and voltage sensors; mechanical or electrical controllers; actuators; plus valves; switches; microprocessors; and many other components.

#### 3.2.2. The Gearbox

The main rotating machinery in a typical HAWT is the gearbox, which is also the element most prone to mechanical faults. Therefore, it is the component of major interest for Condition Monitoring. 

Gearboxes in HAWTs are generally multi-stage, with one or more sequential planetary stages, followed by one or more parallel stages (i.e., helical gears). The rationale is that the speed of the rotor axis (in the order of magnitude of tens of revolutions per minute, rpm, depending on the wind) is not sufficient for the generator to produce electricity cost-efficiently.

In the typical drive train configuration, the gearbox acts as a rotation speed multiplier, connecting a low-rpm, high-torque shaft on the rotor side (known as the input, slow, or main shaft) to a high-speed (output) shaft on the generator side [28]. The support bearings, mechanical brake, and rotating parts of the generator make up the rest of this common configuration. This can increase the revolutions per minute from 12–30 up to 1200–1800. The efficiency of the gearbox is also linked to the lubricant conditions, as will be discussed in the following sections. These refer to both the oil cleanliness in terms of particle content and the oil viscosity (which influences the thickness of the oil film in the gears and bearings).

#### 3.2.3. The Generator

Located behind the gearbox and driven by the high-speed shaft, the electric generator accommodates the mechanical and electrical components needed to convert the incoming rotation into electricity. In comparison to other generators, the ones in use for wind power need to adapt to a fluctuating mechanical power (torque) source. Furthermore, for wind turbines directly or indirectly connected to local or national grids, synchronous or asynchronous alternators are required to produce electrical energy at the grid frequency (generally 50 Hz or 60 Hz, depending on the national standards). Traditionally, squirrel-cage induction machines and synchronous machines have been used for small scale WTs; doubly-fed induction generators are considered the dominant technology for larger models [38]. Other technologies include the permanent-magnet, switched reluctance, and high-temperature superconducting generators. A comparative analysis of the benefits and limitations of some options can be found in Ref. [39].

### 3.3. Incidence and Main Causes of Structural Collapse

In general terms, structural components are much less likely statistically to be subject to damage and failure than mechanical and electrical components. This is shown in detail in the statistics in Figure 7. For the tower, typical failure mechanisms are caused by bolt loosening induced by dynamic loads [40]. However, as can be seen from Figure 7b, WT blades are the structural elements most affected by damage. This is due to their higher complexity, particular materials, and exposition to strong dynamic loads. The static load-carrying elements (tower structure and substructure, plus foundations) are considered to be less damage-prone. Among the many potential causes of their catastrophic collapse, the structure of the metallic tower is subject to all the classic risks of thin-walled shells, e.g., buckling. Even concrete-made towers may suffer total collapse due to damage at the base (at least one case was reported in Germany in 2000 [41]).

During their operational lifespan (commonly 20–25 years), the blades are subject to rapidly changing dynamic loads, which can cause fatigue damage [44,45], plus other environmental conditions such as rain, humidity, and abrasive wind-carried dust or other kinds of particulate matter. Even the simple uninterrupted exposition to sun-radiated ultraviolet rays causes durability issues in the long term. All these eventualities might cause surface damage and/or corrosion.

Moreover, the blades can be hit by larger foreign objects, such as hailstone and bird strikes, and (commonly) by lightning. These two latter factors account (in the same order) for 16% and 20% of the total damage causes according to Ref. [36], making them the most common causes after manufacturing defects. It is estimated that one-third of all damages on blades happens during storm events [46].

The blade tip and trailing edge (refer to Figure 5) are generally designed and shaped to minimize aerodynamic noise, which makes them thin and particularly prone to the occurrence of mechanical damage. A full-scale static test showed that a 40 m-long E-glass/epoxy composite WT blade can bear tip deflections up to 11 m under flap-wise loading before structural collapse [47]. This large flexibility, however, may cause the blade tip to hit the HAWT tower under exceptional wind conditions. Due to its cross-section, the blade is much stiffer edge-wise. In conclusion, seven categories of damage are considered as the most common ones in composite blades; these are reported in Table 1.

### 3.4. The Incidence and Main Causes of Mechanical Failure

Recalling Figure 7b,c, one can see that the mechanical failures (including brakes, gears, drive train, and generator) are preponderant in terms of both total failures and downtime. For this reason, the majority of scientific research focuses on the CM of these components housed inside the nacelle.

In the context of mechanical components, the concept of the “failure rate” is generally applied. The standard time-scheduled maintenance, therefore, should follow the classic bathtub curve, focusing on early-stage monitoring (to assess “infant mortality”, due to defective components) and in the long run, where wear out failures start to become predominant. During its life cycle, any wind turbine is however subject to the random occurrence of unexpected faults; this represents a statistical constant along time. In general, gearbox failures happen slightly more frequently due to late wear out failures compared to early infant mortality [21]. A study over a 13-year timeframe showed a failure rate from 0.10 to 0.15 mechanical failure/turbine/year for land-based European WTs [51]. 

Bearings are, arguably, the most critical component of the gearbox. A 2013 report from the U.S. National Renewable Energy Laboratory (NREL) [52] estimated that about 70% of all gearbox failures are due to bearing failures. More specifically, the bearing life of the parallel stages is generally considered to be very high, in many cases over 1 million hours [20], even if bearing failures at this stage are nevertheless not uncommon (see, e.g., Ref. [53]). On the other hand, bearings in the planetary stages often do not fulfil their design lifespan [20]. This has also been assessed in many experimental campaigns [53] and it is quite understandable due to the large torque applied at the planetary gear stages. The reliability and durability of these bearings have always been considered one of the major issues for WTs [54]. Many failures are observed in the bearing inner rings of the planetary stages since these are exposed to ~5 times more high load cycles than the outer rings [20]. Table 2 reports a brief description of the most common bearing failure typologies, according to the nomenclature reported in Ref. [55].

On the other hand, the gearbox gears can be generally assumed to fulfil their design life as well, even if their reliability is often lower than expected and depends on the manufacturer’s design, tooth profile, and material quality [20]. Surface fatigue cracks and tooth bending fatigue are among the most common causes [20].

Outside of the gearbox, the main shaft bearings are considered to be much less critical and are generally capable of fulfilling their design life [56]. The generator presents some rotating components that require CM as well. This can be performed with vibration-based inspection (VBI) and signal processing approaches (two examples are reported in Refs. [57,58]) or using alternative methods such as temperature trend analysis [59].

In conclusion, Table 3 reports the main causes of mechanical failure according to the scientific literature. However, one should remember that, apart from purely mechanical failures, generators are (obviously) at risk of the failure of electrical materials [60].

### 3.5. Survey and Maintenance Policy for Offshore Wind Farms

Due to their greater logistic complexity and higher O&M costs, a brief discussion about the maintenance strategies for on- and offshore wind farms is needed.

“Corrective” or “reactive” maintenance is, by definition, performed after failure detection, aiming at the restoration of an asset to a condition in which it can perform its intended function. It is both unscheduled and unplanned. Thus, corrective maintenance is often unavoidable, with the maintenance teams having to respond to equipment breakdown or failure immediately with little to no pre-alarm and often without a totally clear understanding of the exact cause of damage/failure. “Preventive” maintenance, on the contrary, aims to carry out an overhaul, replacement, or repair before the component fails, in a pre-planned, predictable fashion. This can help to reduce downtimes while also improving efficient resource planning.

Preventive maintenance can be further divided into pre-determined (time-scheduled) and predictive (condition-based) maintenance. This scheme is the one currently set by the current European standard, as dictated by the norm EN 13306:2017 (depicted in Figure 8). 

The current trend in the wind industry is to transition from time-scheduled to condition-based monitoring, using embedded sensors and continuous, permanent monitoring systems to detect early signs of anomalies at a global level. This is mostly achieved through vibration-based signal processing, SCADA data analysis, and statistical pattern recognition, applying artificial intelligence strategies (mainly machine learning and artificial neural networks). As mentioned, these aspects will be all addressed in future, dedicated works. However, once an anomaly is detected (or even localized) at a global level, on-site maintenance is still required for in-depth, localized analysis. Therefore, local NDE and global vibrational approaches are not mutually exclusive. On the contrary, they both allow fast and cost-efficient maintenance planning. This concept is a key component of the “Intelligent Maintenance” framework, based on a globally optimum trade-off between prevention and repair costs [37].

As already mentioned before, the issue of proper maintenance planning is even more prominent for offshore installations. Table 4 reports the main factors that affect the selection of the maintenance and survey strategy for offshore wind farms, according to selected authors from the recent scientific literature.

## 4. Main Applications for NDE Techniques in Wind Turbines

Non-destructive evaluation strategies and techniques can be applied for SHM, CM, or both. They are used to evaluate the (global or local) structural integrity of the load-bearing and mechanical components, as well as to detect growing cracks, manufacturing defects, or the deleterious consequences of external actions or prolonged operating conditions. However, some specificities depend on the particular system under investigation. For instance, the SHM of foundations (especially deep ones) has its own specific needs and more compelling limitations. WT blades, for economic reasons, are generally made of fibre-reinforced plastic. However, these composite materials are quite prone to develop manufacturing defects and are more difficult to assess in detail than other more conventional building materials (e.g., structural steel). Finally, the drive train and other mechanical components inside the nacelle are subject to a high torque and/or speed rotation and/or temperatures, differently from the wind turbine external structure.

For these reasons, it is possible to define some well-defined fields of application, described hereinafter.

### 4.1. Condition Monitoring of the Mechanical Components

Arguably, this is the most relevant aspect due to the statistical prominence of mechanical faults over structural collapses in WTs, as discussed before. Indeed, it can be considered as the main focus for the whole wind industry [72]. Vibration-based approaches are one of the main strategies in this ambit, even codified by specific requirements (see e.g., ISO 13373-1:2002 and ISO 61400-25-6:2016). SCADA data analysis is also highly estimated for this aim. However, as mentioned earlier, due to the vastness of this specific argument, these approaches are not included here and will be deferred to a dedicated future work. Oil monitoring is another very common approach, while other NDTs, often performed locally, include acoustic emissions, infrared thermography, and electromagnetic testing. These will be all discussed in detail in the next sections.

### 4.2. SHM of the Wind Turbine Blades (Blade Monitoring)

Probably the second most important aspect, again due to the relative frequency of (partial or total) blade failures. To account for all the potential sources of damage enlisted in the previous section, acoustic emissions and strain measurements are commonly applied, both on- or off-site, at different life stages of the blade (quality checks after manufacturing, routine maintenance, extraordinary repairs, and forensic studies on collapsed specimens).

Many studies involve the dynamic and/or static characterisation of the WT blades, off-site or in situ. Of course, the first case generally produces more reliable results, yet it requires detaching the blades from the rotor and carrying them in a laboratory, with all the related costs and delays. On-site inspection, on the other hand, can be performed (depending on the technology) on stationary blades, temporarily halting the energy production, or during operations, without interruptions. 

The National Wind Technology Center (NWTC) of the NREL performed a remarkable full-scale fatigue testing of two 9 m-long CX-100 WT blades (one pristine benchmark and one with purposely inserted defects) [73,74]. Strain gauges, piezoelectric transducers, and accelerometers were used to measure the vibrational response of the two systems. These were analyzed to identify the onset of fatigue injury and to study its progression with a variety of different signal processing techniques.

Generally, many different sensing techniques are jointly applied for a complete assessment; e.g., the European project ReNEWiT developed a lightweight, automated multi-sensor gantry system for the monitoring of stationary WTs (up to 40 m-long) GFRP blades. It included an X-ray Compton backscattering system, dual laser shearography, pulsed thermography, and phased array ultrasonic testing. These (and other) approaches will be presented separately in the rest of this discussion.

Regarding the sensor placement, they can be located close to the blade “hot spots”, i.e., the locations where damage is most likely to occur. These were enlisted in Ref. [50] as: at the blade root (where the mechanical stress is maximized);between 30% and 35% of the chord length;at 70% of the same;at the maximum chord section (subject to potential buckling);on the upper flange of the spar, at different chord lengths depending on the pitch angle (thus on the current wind speed).

### 4.3. SHM of the Supporting Structure and Substructure

While overall a rare occurrence, the structural failure of the tower itself has a high level of risk due to the severity of the potential consequences (total asset loss). In this sense, both global and local monitoring are required. The first case deals with the overall structural integrity and stability; this is performed, generally, with vibration or strain measurements, even if Global Position Systems (GPSs) sensors and inclinometers are commonly used as well to monitor potential risks of subsidence or capsizing, especially offshore. Periodic controls to avoid tower misalignment are required by many regulators as well.

Local monitoring focuses on specific high-risk components. These damage-prone hot spots include: welted, grouted, and bolted joints, due to their relative fragility, in particular to fatigue damage; e.g., on tripod offshore structures, the upper central joint is the most critical location;location exposed to an aggressive environment (e.g., underwater or, even worse, in the splash zone). Corrosion monitoring is particularly requested in these most endangered locations.

Thus, while much less developed than drive train CM, wind turbine SHM is gradually gaining industrial applications as well. For instance, the German BSH standards (in force since 2007 [75]) mandate that at least 1 out of every 10 offshore WTs is equipped with an SHM apparatus dedicated to the support structure.

### 4.4. SHM of the Foundations

The concepts expressed for the load-bearing structure can be extended to the foundations, with the additional risk induced by a higher degree of uncertainty. This derives from both the complexity of soil–structure interactions (especially underwater on the sea bed) and the difficulty to access for periodic checks. Indeed, deep foundations are often needed even onshore, yet they are implicitly inaccessible. They can be assessed via above-ground Remote Sensing (RS) techniques, at the cost of lower reliability, or through lengthy excavation campaigns (when possible), at the cost of suspending wind power generation [76]. Both choices are sub-optimal from a cost-benefit point of view.

For offshore installations, in addition to this problem, scour monitoring is strictly required. In particular, for monopile HAWTs, a scour-induced reduction in the foundation integrity over time may lower the first natural frequency of the support structure, making it dangerously closer to the frequencies where most of the broadband wave and gust energy is contained [77]. Thus, more wave energy may become resonant with the structure, increasing material fatigue [78]. This is even more dangerous in seismic areas due to the potential joint action of seismic and wave loads on monopile structures [79].

Inclinometers, strain gauges, and optical fibre sensors are all commonly used for foundation monitoring. To protect them from the harsh external environment, they can be embedded in the concrete pour (as tested e.g., in Ref. [80] with fibre Bragg gratings). The same sensors can be used to detect surface cracks in shallow foundations as well [76]. Displacement sensors, such as laser or infrared telemeters and tachymeters [81], can be located in the foundation to monitor vertical movements considered as precursors of certain failure mechanisms [82]. 

## 5. Non-Destructive Techniques (NDTs)

In this section, the main NDTs used for HAWT integrity monitoring are reviewed and discussed. Please consider that, while all techniques are reported here separately for better comprehension, they can (and should) be applied synergistically to compensate one another for their limitations. As an example, He et al. [83] proposed combined ultrasound and vibrothermography testing for the damage detection of very small surface damages in composite panels.

For all the tables reported in this and the following section, the following inclusion criteria have been applied. Due to the large range of different aspects, as well as the vastness of the scientific literature on each subtopic, only a selection of relevant, recent, and well-recognized scientific articles is included. 

In more detail, all articles (except wherever specified differently) have been published in the 2000–2021 period and had at least ≥30 mentions as of 1 December 2021. Please consider that the review should not be considered exhaustive, and it is therefore only indicative of the most prominent and recent advancements in each field of research.

For each reported study, its main declared field of application will be indicated as SHM (external structure, blades, foundations) or CM (any high-speed rotating machinery component).

### 5.1. Traditional, Enhanced, and Automatic Visual Inspection (VI)

Simple Visual Inspection is still regarded as the most common form of maintenance survey for both the tower, the exterior of the nacelle, and (especially) the WT blades. This is generally performed from the tower bottom through binoculars or, during blade cleaning or inspection routines, by roped maintenance staff and/or from a gantry or lift system. These latter solutions expose the personnel to a certain level of danger and are not feasible during bad weather conditions. Due to the limited space, the VI inspection of the mechanical components inside the nacelle is more inconvenient, yet still feasible (visual walkarounds). More detailed surveys can be performed thanks to borescope inspection or through the gearboxes’ inspection pots.

The main advantage of VI is that, as a non-contact technique, it is implicitly non-invasive, and thus does not alter the structural conditions of the monitored system (even if a closely-related technique, dye penetrant inspection, applies some external liquid to enhance the visual contrast of surface cracks). On the other hand, it is obviously limited to surface damages and defects alone. Due to the scale of HAWTs, it is rather time-consuming. Even more importantly, it is a qualitative approach; the accuracy of the assessment results varies highly depending on the inspectors׳ skill and is hampered by human errors. These assessments are neither easily comparable between different maintenance teams.

It is possible to classify the VI strategies according to two main factors:if they require human personnel on-site or not;if they add any kind of support to the human eyesight.

The classic visual inspection is generally performed periodically, on-site, by one or more technicians without any form of AI support or advanced device. 

In case I, it is also possible to have maintenance staff members using manned platforms (e.g., small submarines, helicopters, etc.). Otherwise, it is possible to replace the human personnel with (autonomous or not) robot platforms that can fly, swim, crawl, hike, etc., to the needed location. These are essential e.g., in narrow tubes and ducts, or very cost-efficient (when adjusted for risk) for the inspection of components located underwater or at dangerous heights. Several examples of potential designs for a climbing robot can be found in Ref. [84]. Finally, it is possible to deploy a permanent system of closed-circuit cameras for a constant VI (as proposed e.g., with a computer-controlled pan/tilt zoom camera system in Ref. [85]).

Case II is linked to the use or not of computer vision approaches. Generally, survey teams are commonly equipped with hand-held devices (e.g., digital cameras). However, these are often intended for visual documentation rather than quantitative data analysis. These and other optical instruments can both enhance human eyesight or replace it completely. Two examples of the first case are the use of the line or edge detection methods such as the ones applied for WT blade surface crack detection in, respectively, Ref. [86] and Ref. [87]. In this sense, the Canny algorithm was found to be the most efficient choice in many similar studies [88,89]. Similar techniques were utilized to estimate the extension of the surface area damaged by machining processes (e.g., drilling) in composite laminates for WT blades [90].

In this second case, the term automated visual inspection is used. Automatic computer vision approaches can be image- or video-based. These allow for the inspection of large surfaces rapidly and reliably, even for the smallest surface defects, including e.g., barely visible impact damages (BVIDs) due to bird strikes on composite materials. 

To summarize, non-conventional VI strategies can be classified (as in Table 5) depending on the (manned or unmanned) platform and the potential use of any Computer Vision techniques.

Regarding the use of robotized inspection, unmanned aerial vehicles (UAVs) are the predominant choice. Other non-flying alternatives are more uncommon and have received less attention from practitioners and researchers alike. For instance, Lim et al. [100] proposed the concept of an inchworm-like robot, with telescopic motion, for blade VI. The concept is intended to carry several sensors, including a camera. Damage detection and localisation are achieved for surface cracks using computer vision and blob labelling.

Regarding computer vision and automatic VI, the most common approach is to use deep neural networks (DNNs), especially some sort of region-based convolutional neural networks such as the Fast R-CNN [101], faster R-CNN [96], or similar variants, to automatically detect, localize, and estimate the severity of surface cracks. Often, reliable R-CNNs architectures, pre-trained over some generic image dataset, are employed after transfer learning, fine-tuning, and/or retraining with specific datasets, many of which are already available online. Depending on the typologies of damage included in the training dataset, AI can assess different damages such as cracks, breakages, oil stains, etc. [102] Some recently-developed and fast-growing pre-trained models include VGG-16 [103], ResNet-v2 [104], Inceptionv4 [105], YOLOv5 [106], and EfficientNet [107]. Several variants of these models are available, including hybrid architectures such as e.g., Inception-ResNet-v2 [105]. One recent example can be found in Ref. [108].

However, VI is not the only non-contact approach for NDT. Other RS techniques include, for instance, optical measurement techniques, laser-based approaches, and video spectroscopy.

### 5.2. Optical Methods

Image and video processing are wide research fields. Some optical measurement technologies are well-established and extensively used; the classic example is digital image correlation (DIC), which is well-established for the experimental testing of materials in general [109] and nowadays is quite common for the full-field strain inspection of WT blades in particular (see e.g., Refs. [110,111]; other highly cited works are reported in Table 6). If multiple cameras are available, stereophotogrammetry can be used for strain measurements as well [112]. The method has been tested as well for drone-borne images taken by a multi-copter UAV [113].

Apart from blade monitoring, DIC measurements have been (less commonly) applied to the tower structure as well (e.g., in Ref. [114]). Instead, no relevant applications for CM were found during this literature review.

A more recent video processing technique, the phase-based motion magnification (PBMM [115]), is also gaining researchers’ attention as a means to extract modal parameters from imperceptible vibrations [116,117]. This technique has been proven to be feasible for SHM purposes [88,89] and applied in combination with stereophotogrammetry to WT blades [118]. All these techniques resort, in some ways, to the pixels’ brightness (amplitude) and/or phase and are therefore generally limited to the visible band of the electromagnetic spectrum. They can be preferably used off-site under controlled laboratory conditions (illumination etc.) but have been validated for outdoor investigation on-site as well, with natural illumination.

**Table 6 sensors-22-01627-t006:** Some notable and recent examples of DIC and other optical techniques applied for the NDE of wind turbines.

Study	Year	Technique	Notes	Application
Baqersad et al. [119]	2012	3D DIC	The authors used two stereoscopic high-speed cameras to record the vibrations of a WT blade with optical targets attached to its surface (excited with hammer hits).	SHM
LeBlanc et al. [120]	2013	3D DIC	The full-field displacement and strain fields of one CX-100 9 m-long WT blade were estimated. The damaged areas were located from discontinuities in the curvature shapes.	SHM
Winstroth et al. [121]	2014	3D DIC and point tracking	A random black-and-white dot pattern was applied at four different radial positions on one blade of a three-bladed rotor. The tests were performed in situ on the operating HAWT.	SHM
Carr et al. [122]	2016	DIC and 3D Dynamic Point Tracking (3DPT)	The authors compared the dynamic stress and strain fields obtained with their video-extracted measurements with the readings from attached strain gauges.	SHM

Another optical method is optical coherence tomography (OCT), firstly proposed for the noninvasive cross-sectional imaging of biological tissues [123] and then further refined for high depth resolution in Ref. [124]. The method is based on an external broad bandwidth light source, a charge-coupled device camera, and a series of other components (the setup change accordingly to the specific implementation, several variants exist). The technique has been proven to be suitable for the ultrahigh-resolution imaging of internal defects in fibre-reinforced polymers such as the ones currently used for WT blade manufacturing (CFRP and GFRP) [125]. For these materials, the method was compared to X-ray scanning in [126]. Very recently, Ref. [127] used near- and mid-infrared ultrahigh-resolution OCT for subsurface defects on metal samples covered with marine coatings. This application is well-suited for the undersea parts of the substructure of an offshore HAWT.

One highly cited work (>30 citations as of 1 December 2021) was found in the scientific literature for OCT; Liu et al. [128] studied the delamination growth in a GFRP WT blade. A tridimensional geometric model of the crack surfaces was reconstructed.

### 5.3. Laser-Based Measurement Techniques (LDV, LiDAR, and Shearography)

Some applications of laser technology in HAWT monitoring include laser vibrometry and laser scanning [129]. However, apart from the specific acquisition techniques, 1D, 2D, or 3D laser Doppler velocimeters (LDVs) will return velocity time series that can be used for vibration-based SHM, which will be discussed elsewhere. Another example of laser-based approaches is the light detection and ranging (LiDAR) technology, which was proposed e.g., in Schäfer et al. [130] to be mounted on a multicopter UAV prototype for the 3D mapping of HAWTs in situ. 

Finally, shearography testing (ST), is another optical technique that applies coherent laser illumination for surface deformation measurements. It can be considered as a sort of speckle interferometry with laser point patterns; thus, it is a short-range laser-based NDT. ST offers full-field and fast (even real-time) non-contact imaging; it can then be used for defect and damage detection by searching for deformation anomalies [131]. The technique has been investigated since the early 1990s [132], yet is used mainly for indoor industrial applications or under controlled laboratory conditions only (e.g., Ref. [133]), even if portable devices for in situ testing are nowadays available. However, this equipment is generally more expensive and complex than the ones for other NDTs [133].

Shearography has been applied to detect delamination, debonding, impact damage, wrinkles, and dry spots in composite WT blades [131]; to monitor their response under quasi-static failure tests [132]; to evaluate the soundness of bonding in laminated composites [134]; and to assess bird strike BVID in carbon/glass fibre reinforced plastic sandwich panels [135]. Its results have been compared to the ones from active infrared thermography (in Refs. [136,137]) and the ones from image correlation, acoustic emission, fibre-optic strain sensing, and piezoelectric sensing (in Ref. [138]). From these and other studies, it has emerged that the ST technology may still require future research [139]. Nevertheless, some related studies have been published very recently (e.g, Ref. [140]), even including remotely-controlled robotic platforms [141]. 

### 5.4. Video Spectroscopy

Even not considering laser-based approaches, optical methods are not strictly limited to the visible portion of the electromagnetic spectrum (i.e., wavelengths of 380–700 nm, see Figure 9). The Near-Infrared (NIR, ~700–~900 nm), as well as the Short-, Medium-, and Long-Wavelength infrared ranges (SWIR, ~900–~1700 nm, MWIR, ~3000–~5000 nm, and LWIR, ~8000–~14000 nm, respectively) have all been successfully applied for RS in several engineering applications. Furthermore, the combined use of visible and NIR-SWIR-MWIR can be easily achieved by combining standard cameras with multi- or hyper-spectral sensors and thermographic cameras. For instance, the sensor payload proposed in Ref. [142] for a low altitude land survey was intended to allow for synchronous acquisitions over a band of more than $10^{4}$ nm, with overlaps between the different sensors. This presents several practical advantages since visible, NIR, and SWIR electromagnetic emissions are mainly reflected radiations, while MWIR and LWIR are mostly emitted from the object itself. The former group can provide information regarding the chemical composition of the irradiated object, according to its absorbed bands, while the latter depend instead on its thermal state and energy content. For these reasons, thermographic cameras will be discussed on their own in a dedicated subsection. Short and very short wavelengths (X-rays and gamma rays) will be treated separately as well.

Regarding multi- and hyper-spectral imaging, the difference between the two terms lies solely in the resolution in terms of wavelength bands. Multispectral imaging considers a small number (typically 3 to 15) of spectral bands, while commercially available hyperspectral sensors can discern 244 bands in the VNIR range (sampling 2.3 nm per band) and 254 in the SWIR one (5.8 nm per band) [143]. In both cases, the spectral signature of each material can be used for supervised or unsupervised classification; many algorithms have been developed for this aim.

This research field is, however, still underdeveloped for WT applications. Rizk et al. [144] discussed the potential offered by a hyperspectral imaging system in detecting damage to WT blades and icing events. Several damage typologies were considered; the results demonstrated that hyperspectral imaging could detect surface and subsurface defects, as well as icing events in their early onset stages. A similar approach was then further tested for blade defect detection [145].

Finally, considering even longer wavelengths (>1 mm), microwave sensors have been validated for remote sensing testing, as will be discussed later in a dedicated section.

### 5.5. Infrared Thermography (IRT) and Other Temperature Measurements

Infrared thermography, also known as thermal imaging, is one of the most common optical techniques. Indeed, it is applied extensively both for SHM purposes to the external structure and blades and for the CM of the rotating machinery inside the nacelle. 

The IR sensors can be single-point transmitters or cameras with different levels of spatial definition (generally in the order of the hundreds or thousands of pixels). Many commercial cameras offer a dual view (visible and IR) simultaneous recording, both with a pan/tilt control or fixed angle of view. They can also be deployed in parallel and directly connected to a SCADA system. They do not need to be in direct contact with the target; this is especially useful for hot machines. However, normally, the sensor must be mounted to a close distance from the object to obtain reliable readings. Because of this requirement, it is very rare to temperature monitor more than one target with a single IR transmitter/camera. Due to the implicit line-of-sight limitations, IRT is generally limited to a single surface; however, 3D scanning and computer-aided model reconstruction also allow for 3D thermography over all the external surfaces of a single object. This was proven to be feasible for WT blades in Ref. [146].

#### 5.5.1. Passive IRT

Inside the nacelle (indoor), temperature readings are particularly useful for the CM of the gearbox, mechanical brake, generator, main shaft bearing, the yaw and pitch systems, and the pump motor of the hydraulic system. In Ref. [147], it was suggested that any temperature rise between +1 and +10 °C should be considered as a potential index of minor damage. The same guidelines suggest planning for a repair in 2 to 4 weeks for an increase between +10 and +35 °C, in 1–2 days for +35–+75 °C, and immediately if higher than +75 °C (all these indications must be corrected according to the environmental and operating conditions). Thermal images can be used for supervised learning, training a classifier with readings from the healthy conditions as completed e.g., for brushless DC motors in Ref. [148]. Furthermore, these can be used for the detection of non-mechanical failures such as fire detection and the monitoring of the high voltage transformer and other electrical systems (power electronics, control system, etc.). All these uses are well-described in Refs. [149,150].

Apart from damage detection, IRT is especially well-established outdoor for ice detection on WT blades (see e.g., Ref. [151]). This is relevant from an SHM perspective since freezing conditions are known to increase the stiffness of the structure [152] and therefore can change the vibrational response of the same. In turn, this might cause false alarms for vibration-based anomaly detection, if these confounding influences (i.e., damage-unrelated environmental effects) are not actively depurated from the recorded output. For temperature monitoring under operating conditions, Ref. [153] recently proposed a line laser thermography approach, reading a fixed point along the rotating blade. IRT can even be used to monitor the blade de-icing systems (generally an embedded heating wire) for electrical breakdowns [154].

#### 5.5.2. Active IRT

Indoor and outdoor IRT are not limited to passive thermography. Active heating and cooling thermography techniques are available for different purposes; a large set of options were investigated on WT blade samples in Ref. [155]. However, these approaches need some sort of controllable thermal excitation. This can be induced through, e.g., thermal emitters, ultrasounds, microwaves, eddy currents, flash lamps, etc. 

In the case of surface heating (flash lamps [156], lasers [157], etc.), the properties of surface defects (such as their depth [158]) can be estimated based on the heat conduction from the surface inward. This approach is known as surface heating thermography (SHT). It has been also referred to as optical thermography [159] and it is (generally) performed in reflection mode, i.e., with the IR camera and the source heat on the same side of the target surface. Pulsed, pulsed phase, stepped, modulated, lock-in, and line scanning thermography are all feasible within SHT.

On the other hand, microwaves, ultrasounds, and high-frequency induction currents can be used for volume heating thermography (VHT), i.e., heating inside out. These applications are also known as non-optical thermography. A more detailed discussion about optical and non-optical excitation sources can be found in Ref. [160].

According to the energy source, it is possible to further classify the VHT techniques as:eddy current (EC) thermography (or inductive thermography), based on the heath released by resistive losses to the eddy currents induced by electromagnetic pulses [161];microwave thermography, based on the well-known principles of microwave heating. The electromagnetic energy is absorbed volumetrically by the target object, favouring uniform and rapid self-heating;vibrothermography (or thermo-sonic testing), with mechanical waves.

Note that, due to their low electrical conductivity and magnetic permeability, fibre-reinforced polymers have a high penetration depth (about 50 mm for CFRP under 100 kHz excitation) and thus can be volumetrically heated with ECs [162]. 

Some VHT approaches include EC pulsed [163], pulsed phase [162,164], stepped [165], and lock-in [166] thermography, as well as microwave pulsed and lock-in thermography [167]. Most of these strategies can be performed with ultrasonic waves too; another less widespread alternative is the ultrasound-burst-phase thermography [168]. The links between these VHT techniques and their SHT counterparts, mentioned before, are graphically displayed in Figure 10.

For completeness’ sake, the main aspects of the most prominent active IRT techniques will be recalled in the next paragraph, considering both surface and volume heat sources.

As the name suggests, pulsed thermography is based on short-duration energy pulses, while the pulsed phase is based on a phase analysis in the frequency domain and thermal wave conduction [169]. These two techniques are, arguably, the most common alternatives. The former can be used to detect several damage typologies, e.g., air bubbles [170], inclusions of foreign matter [171], deficiency in adhesive bonding [170], and other glue faults [172] in GFRP WT blades. The latter was tested and validated, again on GFRP WT blades, with flash lamps for deep defects and delaminations [173]. 

Regarding non-optical pulsed approaches, EC-based pulsed thermography has shown several successful applications in the last decades, especially for CFRP materials, e.g., for delamination depth evaluation [174], impact damage [175,176], damage classification [177], etc. However, it remains limited by technical issues such as the lift-off effects [178] and the difficulty to discern subsurface defects in the inner layers from debonding at outer layers [179]. EC pulsed phase thermography was tested on CFRP laminates in Refs. [180,181]. Microwave pulsed thermography was tested on both CFRP specimens (with artificially-added damages in Ref. [182], for delamination detection in Ref. [183], and for comparison to laser-based pulsed thermography in [184]) and GFRP [185].

For what concerns the other (non-pulsed) active IRT techniques, stepped thermography uses discrete increases in the temperature, while modulated thermography uses frequency-modulated thermal waves. Lock-in thermographic analysis uses periodic input energy waves to detect and localize internal inhomogeneities. This is achieved thanks to the interferences between incoming and reflected heat waves; thus, the strategy is similar to ultrasonic testing, as will be discussed later. The technique was successfully tested with both optical and non-optical heat sources. In Ref. [186], 1400 W halogen lamps were utilized for the detection of skin–skin and skin–core delamination in a CX-100 WT blade. A commercial microwave over was used in Ref. [187] with a CFRP specimen. Lock-in vibrothermography was tested for delamination detection of CFRP specimens in Ref. [188]. Vibrothermography can be also performed with high-energy bursts in the 10–50 kHz range [189].

Finally, line scanning thermography is a NASA patented technique that uses an IR sensor moving in synchronous with the heat source to dynamically investigate metallic or composite surfaces. It has been tested for impact damages on CFRP panels [190] and to assess manufacturing defects in GFRP blades [191].

Some relevant studies for both active and passive IRT are reported in Table 7. Most of the scientific articles encountered in this review discussed SHM applications for WT blades; only a few works discussed applications for rotating machinery components such as, e.g., fatigue-damaged gears [192]. Several of these techniques can be applied to metallic/ferromagnetic components as well, for the structural monitoring of HAWT towers (e.g., using pulsed EC to detect corrosion [192]).

#### 5.5.3. Physically-Attached Temperature Sensors

While conventional or advanced IRT are more prominent for blade inspection, temperature changes are often utilized for monitoring rotating machinery components as well. However, this is generally achieved not by RS but rather by employing physicallyattached sensors, such as thermocouples or similar electrical devices. Usually, at least three temperature sensors are installed at the main shaft support bearing and the high-speed shaft bearing, and for the lubricant oil [205]. The operating temperatures of the generator, converter, and transformer are often monitored as well. These readings are generally included in the SCADA dataset along with ambient temperature [206]; therefore, they are generally processed along with several other thermophysical measurements to detect anomalies. Some noteworthy examples for CM focused solely and/or prominently on temperature readings can be found in Guo et al. [59], Guo & Bai [207], Cambron et al. [208], and Astolfi et al. [209]. Even more specifically, approaches based on oil temperature measurements were recently reviewed by Touret et al. [210]. 

### 5.6. Radiographic Testing (RT)

Radiography is a conventional approach for the internal inspection of structures and mechanical systems. In this regard, the most common radioactive sources available for industrial applications are X- and Gamma-rays. However, the distinction between X-rays and Gamma rays is not always very clearly defined. Depending on the conventions, wavelengths between $\sim 10^{-10}$ and $\sim 10^{-12}$ m are generally considered X-rays. Shorter wavelengths are classified as Gamma rays. Neutron radiography is also a viable application, even if there is a very limited number of available examples encountered in the scientific literature (e.g., in Ref. [211]). 

Computed Tomography (CT), specifically, saw a relevant increase in popularity in the last decade, mainly due to the increased availability of X-rays, improvements in spatial resolution, and the reduced acquisition time [212]. These CT scanners can be 2D or 3D and portable (hand-held or via manned or unmanned platforms such as in Ref. [213]), even if the largest ones are fixed and available at specific laboratory testing locations. These allow sections of WT blades up to 4.5 m-long to be scanned in a single take [214].

The main concept is to transmit ionising radiation throughout a dense material, measuring its attenuation along the photon path. For a homogeneous material, the amount of total attenuation would be a spatial constant, only depending on the thickness and density of the material. This is visually presented as pixel readouts, generally in a greyscale. Flaws and density inhomogeneities can be detected and located as anomalies in the image outputs, with extremely good spatial resolution thanks to the very short wavelengths utilized. This technology is particularly efficient for voids and other discontinuities that lay parallel to the ray beam. On the other hand, the energy of photons is proportional to their frequency (thus, inversely proportional to wavelength). This makes RT one of the most energy-demanding NDTs. Therefore, it is a relatively expensive technology, and harmful for biological tissue (since the rays can interfere and damage them at the molecular scale). Thus, the operators must be adequately protected and the procedure is inherently more hazardous than other options.

For this and other practical reasons, RT is more common for laboratory experiments rather than in situ applications. Thus, for wind turbine applications, RT can be better used for material testing, research on fracture mechanics, and forensic analysis after structural failure. For instance, Mishnaevsky Jr. et al. [215] used an X-ray CT to investigate the erosion mechanisms on the leading edge of WT blades at a microscopic scale. Jespersen and Mikkelsen [216] and Baran et al. [217] used the same technique to investigate, respectively, the evolution of fatigue damage and manufacturing defects (fibre misalignment and porosities) in GFRP specimens intended for blade manufacturing. For the gearbox and other mechanical components, an X-ray CT was used on laboratory experiments by Gould et al. [218] to map the distribution of White Etching Cracks (WECs) networks within failed bearings, to assess the effects of subsurface steel inclusions as initiation sites. Gegner & Nierlich [219] used an X-ray diffraction-based residual stress analysis on operating WT gearbox bearings to investigate the effects of vibration loading on WECs.

### 5.7. Microwave and Terahertz Testing

Microwave and terahertz (THz) testing are based on electromagnetic radiation with long and very long wavelengths, ranging from 1 mm up to 1 m and beyond. These correspond to frequencies between 100 MHz and100 GHz, for standard microwaves, or 100 GHz–10,000 GHz (0.1–10 THz) for THz approaches. Differently from the short-wavelength wave-based approaches seen before (and from ultrasonic tests, which will be discussed later), microwaves can penetrate dielectric materials and thus interact with their inner structure with limited signal attenuation [159]. THz waves can penetrate even thicker layers of low- and very-low-conductive polymers such as GFRP and other fibre-reinforced polymers used for the manufacturing of WT blades (except for CFRP, which is instead too conductive to be inspectable with THz testing [220]).

Indeed, both microwaves and THz techniques are particularly applied for blade monitoring. Indeed, the only scientific article of this group that met the selection criteria (>30 citations as of 1 December 2021) to be considered highly influential was the work of Kuei Hsu et al. [221]. They used time-domain spectroscopy and terahertz radiation to detect damages inserted by sawing small cuts into GFRP laminates (intended for the manufacturing of WT blades). Martin et al. [222] compared a terahertz inverse synthetic aperture radar system with X-ray and IRT imaging for the inspection of GFRP WT blades’ spar caps.

Some other recent applications to GFRP WT blades—which also include THz-inducted active thermography—are reported in Refs. [223,224]. Applications for defect detection on both CFRP and GFRP composites are discussed in Refs. [225,226]. Im et al. [227] reported an application of THz testing to their trailing edge. On the other hand, there does not yet seem to be any relevant study on the microwave or THz inspection technologies for the condition monitoring of WT rotating machinery elements.

For HAWT towers, microwave technologies can be used to inspect the metallic surfaces when covered with epoxy or other insulating paints, as it is often applied for long-term steel protection in severely corrosive atmospheric conditions. This has been proven to be feasible in Ref. [228] for fire protect-coated steel panels, using frequencies between 8 and 12 GHz. THz waves were also proved to be able to detect corroded metals under thick insulating layers, plus water intrusion in sandwich panels [229]. THs imaging was proven to provide a higher resolution than, e.g., ultrasound testing; on the other hand, it is limited by a lower penetration capability [230].

Considering a different approach, Pieraccini et al. [231] used a portable, high-speed continuous-wave step-frequency interferometric radar, which transmits continuous microwaves (central frequency: 16.75 GHz) at discrete frequency values, to remotely record the dynamic behaviour of an onshore WT.

### 5.8. Electromagnetic Testing (ET)

Electromagnetic Testing, especially using the already cited ECs, relies on the use of changes in the electric conductivity to detect and localize damages in metallic and non-metallic components. A complete review, accounting for several fields of application, can be found in Ref. [232]. This strategy is well-known and widespread in manufacturing industries. Thus, these techniques are viable for the tower structure and substructure. For instance, they are widely used for weld inspection [233]; hence, they can be applied for the inspection of the circumferentially and longitudinally welded connections in the tubular steel components.

EC testing can be used for conductive composites as well, e.g., for CFRP WT blades, enabling the detection of both surface and subsurface damages and defects. However, while the concept has proven to be feasible (see, for instance, Ref. [234]), the relatively low conductivity of these composites might compromise the detection accuracy for certain typologies of damages such as delamination [234], making EC thermography a preferable option.

Pulsed ECs [235] have also been used to detect steel corrosion [236] and low-energy impacts in CFRP composites [237]. The first application is apt for the monitoring of the tower structure and substructure, especially in critical areas such as the splash zone. The second technique can be advantageous for blade inspection e.g., after a bird strike.

Finally, radio frequency EC testing has been suggested for less conductive materials such as CFRP [237], being applied for the detection of fibre misalignment, gaps, and local polymer degradation [238,239].

However, despite the several potential applications, no SHM approach based solely on EC seems to have obtained noteworthy attention from the scientific community. Hence, the preferred use of ECs remains for volume heating and IRT.

Electromagnetism-based strategies are less commonly found for CM, except for some applications of electrostatic monitoring of WT gearboxes [240], often resorting to a single [240,241] or multiple [242] oil-line electrostatic sensors. The concept has been tested for WECs in WT gearboxes as well [243].

### 5.9. Acoustic Emissions (AEs)

The key concept of AE is that, when an internal crack propagates, it releases energy in form of acoustic waves. Debonding, delamination, crushing, and other kinds of damage produce localized, transient changes in the stored elastic energy as well. These elastic waves are therefore also known as stress release waves [50] and travel inside the material. AE testing is based on their detection. This basic procedure is sketched in Figure 11.

These acoustic waves are generally too weak to be heard by a human bystander, yet they can be easily detected from (one or more) sensing devices attached to the surface of the inspected element. There are, however, two main issues: (1)not all the typologies of damage emit strong AE;(2)even more importantly, many damage-unrelated phenomena emit AEs.

Therefore, especially for in situ testing, these confounding influences may exceed crack-related emissions.

Assuming their correlation with damage, once detected, these non-audible emissions can be then also used to estimate and (with multiple sensors) locate the origin of the damage. This last task is generally performed via time-of-flight triangulation, even if alternatives with less than three sensors have been proposed, e.g., in Ref. [244].

In SHM, AE event detection is particularly common for WT blade laboratory testing since AEs occurring during loading conditions are very likely indicative of the presence of crack propagation. In a load-hold test [245], the WT blade is loaded slightly above the highest service load and then held in position for around 10 min. For an undamaged fibre composite structure, AEs will occur only during the first loading and not be re-emitted significantly on subsequent reloading to the same level. Therefore, sustained emission during a load-hold is considered indicative of damage [245]. This was documented as early as in the 1990s during the loadings of blade fatigue tests [246,247].

The most relevant examples of As applications for WT blade monitoring are included in the second part of Table 8; some other less cited, but still noteworthy studies include the research completed by the Centre for Renewable Energy Sources (CRES) on an NM48/750 NEG-MICON WT blade, monitored in-service [248,249].

For CM, the basic concept is that faulty mechanisms (e.g., bearing defects) disrupt the AE waveform, causing a detectable divergence from the readings under normal operating conditions. Historically, these changes in the AE signatures have been considered to be observable earlier than significant alterations in the vibrational signatures of the same pieces of rotating machinery, thus allowing for early damage detection and prognosis [250].

AE techniques have been proved feasible for the monitoring of ball bearings, standard roller bearings [251], and tapered roller bearings [252]. Soua et al. [253] evaluated their feasibility for gearboxes and generator shafts. Purarjomandlangrudi & Nourbakhsh [254] tested AEs to detect a fault in the outer race bearing in a low-speed shaft rig test, simulating the internal components of a WT drive train.

The first part of Table 8 reports several relevant studies of the last 20 years for CM. A comprehensive review of previous works about the AE-based structural diagnosis of bearing defects, gearbox faults, and pumps can be found in Mba & Rao [255].

**Table 8 sensors-22-01627-t008:** Some notable and recent examples of AE techniques applied for the NDE of wind turbines.

Study	Year	Technique	Notes	Application
Eftekharnejad & Mba [256]	2009	AE waveforms.	Applied for the detection of seeded tooth root cracks in one helical gear of the wind turbine gearbox.	CM
Elforjani & Mba [257]	2010	Continuous AE energy monitoring.	The authors applied AEs for the CM of low-speed shafts and bearings (separately) also considering different conditions such as lubricant starvation. The bearing test demonstrated the AE’s efficiency in detecting crack initiation and propagation.	CM
Eftekharnejad et al. [258]	2011	Kurtogram (spectral kurtosis).	Compared the effectiveness of applying the kurtogram to AEs and for a roller bearing on a laboratory test bench.	CM
Qu et al. [259]	2012	Time synchronous averaging (TSA) and kurtosis.	The heterodyne technique used in telecommunication was used to pre-process AE signals, reducing the sampling frequency from MHz to kHz.	CM
Niknam et al. [260]	2013	PAC-energy (Physical Acoustic Corporation PCI-2 AE system).	This study focused on wind turbine drive trains subject to rotor unbalances. These unbalances may be caused by manufacturing defects or non-uniform accumulation of ice, dust, moisture, or even damage on rotor blades.	CM
Ferrando Chacon et al. [261]	2016	Root Mean Square Error, Peak Value, Crest Factor, and Information Entropy of AE waveforms.	The confounding influences induced by different operating conditions (load and torque) on the AE signature of a wind turbine gearbox were investigated.	CM
Zhang et al. [262]	2017	Damage localisation was performed via triangulation (delays in the time of arrival).	The first attempt of mechanical fault localisation for CM inside a wind turbine gearbox.	CM
Joosse et al. [245]	2002	Load-hold test.	An early application of AEs off-site on a detached WT blade.	SHM
Anastassopoulos et al. [263]	2002	Load-hold test.	Machine Learning (specifically, Unsupervised Pattern Recognition) was applied to AE data from ten WT blades.	SHM
Blanch & Dutton [264]	2003	Load-hold, stationary, and operating tests.	AEs applied on-site to attached blades (both stationary and rotating during normal operating conditions).	SHM
Paquette et al. [265]	2007	Three-point bending test.	The article documented a 5-year long project performed at Sandia National Laboratories (USA) to characterize WT blades made of carbon fibres.	SHM
Zarouchas & Van Hemelrijck [266]	2011	Peak frequency analysis of AEs and Digital Image Correlation.	AEs were used to characterize the crack growth at different scales in laboratory specimens, treated with an adhesive used for WT blades composites. Tensile and compression tests were executed. DIC was used to compare the strain measurements with the recorded acoustic activity.	SHM
Han et al. [267]	2013	Static loading test.	AEs and strain measurements of a WT blade inner shear web were compared, to correlate acoustic emissions and stress conditions.	SHM
Bouzid et al. [268]	2014	Ambient excitation(naturally occurring AEs).	Proposed a Wireless Sensor Network (WSN) architecture for damage localisation in the blades of operating wind turbines (via triangulation).	SHM
Tang et al. [269]	2016	Pencil lead break test.	The acoustic emissions were generated by breaking a pencil lead in the blade surface. Proved the feasibility of damage severity assessment and growth tracking.	SHM
Gómez Muñoz & García Márquez [270]	2016	Pencil lead break test. Damage localisation was performed via triangulation (delays in the time of arrival).	Three macro-fibre composite transducers were applied on the surface of a WT blade.	SHM
Tang et al. [271]	2017	21-day long fatigue test.	Unsupervised Pattern Recognition was applied to a very large dataset of recorded AEs.	SHM

### 5.10. Ultrasonic Testing (UT)

Differently from AE testing, where the acoustic waves naturally originate at the damage location and propagate freely throughout the whole structure, in UT high-frequency waves are generated at a damage-unrelated, user-defined origin. The centre frequency is generally included between 0.1 and 15 MHz but can reach up to 50 MHz; yet, a variety of frequencies can be used, allowing optimization for resolution and/or penetration. As for any portable device, the UT can be performed by manned or unmanned platforms, such as climbing robots or others. Skaga [272] investigated the feasibility of UAV-carried UT sensors, testing it for WT blade monitoring and comparing the results with hand-held instrumentation. 

The source of the ultrasonic waves can be either in contact or not with the target surface. For contact techniques, most of the conventional transducers are piezoelectric; thus, they require a wet film (gel, oil, or water) as a couplant between them and the test object. Water immersion is used for this aim in laboratory testing when the test materials allow it. Nevertheless, some technologies, such as electromagnetic acoustic transducers, do not require the use of a couplant. Silicon membranes or other solid couplants can be used for dry coupling as well, even if they are often limited in the range of frequencies that can transmit. Air-coupled and laser-borne ultrasonic tests are other viable options for water-incompatible materials. However, the air has a very low acoustic impedance, so only a very limited amount of acoustic energy is transmitted. The use of high power pulse lasers (such as Q-switched Nd: YAG and CO_2_ lasers [159]) bypasses this issue while also providing long-range capabilities.

Once emitted, the ultrasonic waves then travel along a well-defined direction through the material thickness or along its surface (guided wave ultrasonic testing), even over long distances. In this latter case, generally, slightly lower frequencies are applied (10 kHz–1 MHz). This allows a much larger inspected area to be covered, also known as the insonified portion of the tested system. Simple transducers (inclined through a plexiglass wedge) or ring transducers can be used. Nevertheless, guided waves are slightly less common than other techniques for wind turbine applications; some applications for NDT can be found in Refs. [273,274]. They are often applied for in-service inspection as they can cover a larger area at once than other UT methods. They have also been recently proposed for ice monitoring on WT blades [275]. 

The working principles of conventional and guided wave UT technologies are represented in Figure 12. Both can be used in reflection mode (with one transducer, analysing the ultrasound echoes) or transmission mode (with one transmitter and one or more receivers). The key concept is that when the (volume or surface) wave interacts with an inhomogeneity, it is partially reflected backwards and partially transmitted forward, with the transmitted wave being attenuated in its amplitude and delayed in its phase (due to the detour around the inhomogeneous area or the different propagation velocity inside it). These differences in amplitude and phase are also quantitatively related to the damage extension [276].

Conventional UT can be applied to a wide range of materials. For the tower structure (and substructure, for offshore installations), it can be used for thickness measurement, monitoring the risk of material loss due to corrosion. These techniques are even more important for WT blades due to their composite materials. Indeed, ultrasound waves are particularly well-suited for fibre-reinforced panels, as the random distribution of the components in the matrix requires high spatial resolution. However, the presence of such fibres leads to sound scattering and directional (anisotropic) damping in GFRP, CFRP, and similar materials [277]. The ultrasonic must travel through several centimetres of these fibre-reinforced polymers; thus, a high voltage ultrasonic pulse is needed to send enough energy into the material. For cross-ply CFRP plates, the effects of lamination and anisotropy must be considered for damage localisation [278].

Apart from the already-mentioned air-coupled and laser ultrasonics, conventional UT can be further classified into many categories. These include pulse-echo UT, phased/linear array UT, local resonance spectroscopy, and others.

In a pulse–echo test, a short-duration ultrasonic pulse is sent into the test specimen using an ultrasonic transducer. The waves then travel through the specimen and reflect at the opposite end of the material, in the absence of inhomogeneities, or at cavities (flaws) and/or discontinuities (delaminations) within the material, if present. The reflected waves are recorded using the same transducer (in sensor mode). The difference in the travelled distances results in different times of arrivals (also known as time-of-flight), allowing the defect or damage to be located through the thickness of the inspected structure. The amplitude and other waveform characteristics can be used as well for further characterisations. 

Phased array ultrasonic testing (PAUT) is a growing and very promising technology. By using many small ultrasonic transducers, each one pulsed independently, it can produce a quasi-flat ultrasonic beam. This can be steered electronically by changing the time delay between the transducers, thus allowing different angles to be inspected without physically moving or turning the portable device. A comparison of advantages compared to standard UT can be found in Ref. [279]. For wind turbine applications, it has been applied for blade monitoring by the researchers of Sandia National Laboratories [280], Lamarre [281], and Zhang et al. [282]. However, PAUT is still under development, and several improvements have been achieved recently (see, for instance, Ref. [283]).

For local ultrasonic resonance spectroscopy, a portion of a large component (e.g., a WT blade) is excited with ultrasounds in a broadband frequency range. The vibrational response is recorded with a nearby sensor and is used to obtain the local material, geometrical, and mechanical properties, for each of the scanning-grid locations [284]. This can be seen as the ultrasound equivalent of the classic hammer and tap tests used for standard local resonance spectroscopy. This NDT is also viable for nonlinear UT e.g., in presence of breathing cracks [285].

Other options can be found in the scientific literature. Ultrasonic pulse velocity was proposed in Ref. [286], specifically intended for monopile HAWT foundations monitoring in combination with other NDE techniques. Single-sided inspection via air-coupled UT guided Rayleigh waves was proposed in Ref. [287] to detect waviness in WT blades. Table 9 reports the main applications found for the timeframe of interest (2000–2021). These are mostly limited to blade inspection, off- or on-site. UT seems to be rarely considered for CM; one of the very few mentions can be found in a very recent PhD thesis [288] for WT gearbox bearings.

### 5.11. Oil Monitoring

Since the early 2000s, oil monitoring and lubricant contamination analyses became very common techniques for machine condition monitoring, also for WTs. Indeed, their usefulness is twofold. On the one hand, it is important to assess the quality of the oil, to prevent mechanical failure due to e.g., lubricant scarcity or solid intrusions (mainly iron or soot particle contamination). On the other hand, several lubricant parameters are considered indicative of the potential occurrence of mechanical faults (see e.g., the previous Table 2). These parameters include the oil viscosity (at +40 °C and +100 °C), potential water content, wear particles in parts per million (or mg/L), the presence of dissolved solvents or gases in the lubricant, and the oil acidity/alkalinity (for potential oxidation) [301]. They can be used for the quantitative assessment of the health conditions of the WT gearbox, hydraulic system, compressor, etc. [301]. Specifically, the presence of wear debris formed in rubbing helps to detect and estimate the severity of wear mechanisms. Total oil contamination can give general information on oil lubricity [302]. Surface fatigue damage of bearing and gear rolling elements, bearing spalling, and gear teeth pitting are typical mechanical failures of WT gearboxes that result in the release of metallic debris particles in the oil lubrication system [303]. In this regard, oil monitoring is quite well developed and codified; e.g., ISO 4406 provides proper thresholds for solid particle content according to their size distribution ((≥4 μm/≥6 μm/≥14 μm). Table 10 reports the most influential publications in the field of WT oil monitoring from the last 20 years.

The scientific literature also reports several examples of sensors developed for oil monitoring. These cannot be reported here in full due to space concerns; one can refer to Hamilton & Quail [304] for a dedicated overview. Only as an example, a couple of noteworthy publications may be recalled. A microacoustic sensor was proposed in [305] for oil viscosity monitoring; Mignani et al. [306] applied wide-range absorption spectroscopy, fluorescence spectroscopy, and scattering measurements to estimate the oil acidity, presence of water infiltrations, and phosphorus content. This latter parameter is considered a proxy of wear since it is generally utilized in anti-wear additives but then absorbed by metallic surfaces over time.

**Table 10 sensors-22-01627-t010:** Some common typologies of oil monitoring strategies encountered in the scientific literature.

Study	Year	Technique	Notes	Application
Myshkin et al. [302]	2003	Optical ferroanalyzer	The document presented the operating principle of the optical ferroanalyzer, a sensing device for the estimation of total lubricant oil contamination, for condition monitoring.	CM
Dupuis [303]	2010	Oil debris monitoring	The technique is based on counting debris particles and measuring their size to assess the severity of the gearbox failure.	CM
Zhu et al. [301]	2013	Several sensing devices	A total of 10 sensors and 6 performance parameters related to oil oxidation, water contamination, and particle contamination were discussed.	CM
Coronado & Kupferschmidt [307]	2014	Water content, particle concentration, particle count, dielectric constant, viscosity, oil colour, and oil density sensors	The paper mainly described a highly accelerated stress screening test chamber to assess the performance of oil properties sensors under extreme ambient temperature and vibration levels. The oil parameters are intended as considered as proxies of wind turbine gearbox conditions.	CM
Zhu et al. [308]	2015	Particle filtering, plus viscosity and dielectric constant sensors	Related to the previous paper by the same authors [301], it applied online oil monitoring for fault detection and remaining useful life prediction.	CM
Sheng [309]	2016	2.5-MW dynamometer test facility at U.S. National Renewable Energy Laboratory (fully described in Ref. [310])	The laboratory tests were performed on full-scale wind turbine gearboxes in three configurations: run-in, healthy, and damaged conditions.	CM

### 5.12. Static Strain Measurements

Static and dynamic strain measurements are both widely utilized for SHM purposes, especially for monitoring the blade static and dynamic response. The specific uses for vibration recordings will not be discussed here; the use of displacement time histories is not different from the one of velocity or acceleration time series. Static strains, on the other hand, can provide insight about the deflection of the WT blades. During laboratory testing, this is useful to characterize the operational and failure behaviour of the specimens. On-site, this allows to both monitor their shape (to avoid collisions with the tower) and to estimate their stress field.

The sensors are generally deployed at the locations of maximum strain, i.e., close to the clamped cross-sections, that is to say, at the blade roots (potentially on both surfaces and directed in both the flap- and the edge-wise directions, totalling four sensing devices) and at the bottom of HAWT towers.

Optical fibres have been extensively investigated and applied for WT strain monitoring. This derives from their several advantages, such as their immunity to electromagnetic interference and their good accuracy. 

In particular, fibre Bragg grating (FBG) sensors are commonly used for strain measurement, especially for WT blades. These can be interrogated with different types of optoelectronic instrumentation. They are advantageous since have the same size and mechanical properties as the original fibre. They can be placed in series, performing many measurements along a single fibre (multiplexing). Deploying many FBG sensors in parallel and perpendicular lines allow, e.g., to detect and track crack growth or to locate an impact. 

Other similar devices include microbend fibres, which were proposed for crack detection in adhesive joints [48,311], and transverse optical fuses, proposed in Ref. [312] for low-energy impact damage detection in laminated panels. The downside is that fibre-optic methods are still an expensive technology and are difficult to implement en masse.

Strain memory alloys have been proposed for strain measurement and shape sensing as well. Verijenko & Verijenko [313] suggested their use for SHM, also for WT blades. For this application, they can be conveniently embedded into the laminate during manufacturing. On the one hand, the scanning for magnetic susceptibility, needed to enable this technique, is labour intensive. On the other hand, these systems can be deployed as actuators as well for aerodynamic load control [314], thus utilized for both tasks at different moments. 

To conclude, some high-impact examples of applications based on strain measurements are reported in Table 11.

### 5.13. Other NDE Approaches

Other less-common NDE strategies and/or applications include:Dynamometer testing, performed off-site on the whole drive train system, to assess for potential slipping behaviour in the high-speed shaft tapered roller bearings [321];Sound-based monitoring, using audio speakers to ensonify the internal cavities of WT blades and arrays of external microphones to detect pattern changes in the airborne sound radiation [322,323];Short-Range Doppler Radar, very recently tested for the son-site SHM of WT blades [324];Multi-sensor apparatuses, such as e.g., the one proposed in Ref. [325] (with optical, acoustical, and vibrational sensing devices) to detect bird and bad strikes.

plus several other techniques that, however, are limited to very few or even only one peer-reviewed scientific articles. These NDE methods are less established than their more classic counterparts reviewed previously. In some cases, this is due to their (relatively) novelty. All these techniques present several interesting qualities but are hampered by specific limitations as well.

## 6. Discussion

Table 12 and Table 13 summarize the main points of this review, reporting the most common application for each NDT (indicated by **X** in Table 12) and their main advantages and limitations (Table 13). This discussion is only limited to the NDT reviewed here; for many applications, please remember that in many cases, vibrational and SCADA data analyses are considered convenient and effective alternatives. These are not included in this discussion and are postponed to a more detailed future work.

Regarding the prevalence of these techniques. as reported in Figure 13 (adapted from Ref. [328], based in turn on data from several sources [329,330,331]), visual inspection remains the most widespread approach. Temperature analysis is also very common, most probably due to the very low cost of physical temperature sensors. UT, AE, and IRT are all (in general) more expensive and thus less frequently deployed. Therefore, it is understandable that the trend is quite clearly decreasing with increasing operational costs. Permanent monitoring apparatuses, on the other hand, represent an outlier relative to this general trend. The reason behind this larger-than-expected deployment despite the higher costs derives from the perceived usefulness of embedded systems. Moreover, the total costs account for both the installation—which is relatively expensive for a global system—and the operating expenses, which instead are relatively convenient. Thus, a vibrational-based, ML-based SHM apparatus becomes more and more convenient throughout the years, when the structure/system as well becomes more prone to damage due to ageing and normal use and consumption. However, as mentioned in Section 3.5, global and continuous monitoring is intended to enable condition-based maintenance. Therefore, it is not an alternative but rather a complement to the local damage assessment capabilities of the NDE approaches reviewed here.

## 7. Conclusions

Despite the great achievements and growth of the wind industry in the last decades, the reliability of wind turbines is challenged by premature mechanical faults, blade failures, and even structural collapses. These issues involve both on- and offshore installations, isolated or grouped in large and dense wind farms. Their consequences are, however, even greater for offshore wind farms, due to the logistic of maintenance in the open sea.

Structural Health Monitoring (SHM) is concerned with the overall load-bearing structure of an asset, while Condition Monitoring (CM) is focused on the fault detection within the subsystems and components of rotating machinery. Therefore, both are essential for the correct functioning of wind turbines.

Here, all the main Non-Destructive Techniques (NDTs) and Evaluation (NDE) strategies for wind turbines SHM and CM have been reviewed and thoroughly discussed. This included the findings of more than 300 documents published in the last 20 years. These covered all the related aspects, for the convenience of both academic researchers and industry practitioners. 

Overall, it is evident that no single option is superior to the others. The synergies of contact and non-contact measurement techniques, especially when applied to the estimation of different physical quantities, should be preferred. Therefore, the main conclusion is that a large set of different sensing techniques can balance out the limitations and drawbacks of each single NDT. This allows detecting damage/fault-related anomalies, as well as discerning them from unrelated operating and environmental variations. This holistic approach can provide a significant economic and safety benefit to the wind industry. 

The NDE approaches reviewed here must be seen in the broader context of Intelligent Maintenance. Thus, their use should be integrated with permanent monitoring apparatuses, embedded in the structure and mechanical components and eventually integrated within a Supervisory Control And Data Acquisition (SCADA) system. These apparatuses, which are becoming more and more widespread and standard in the industry, will be reviewed in future works.

## Figures and Tables

**Figure 1 sensors-22-01627-f001:**
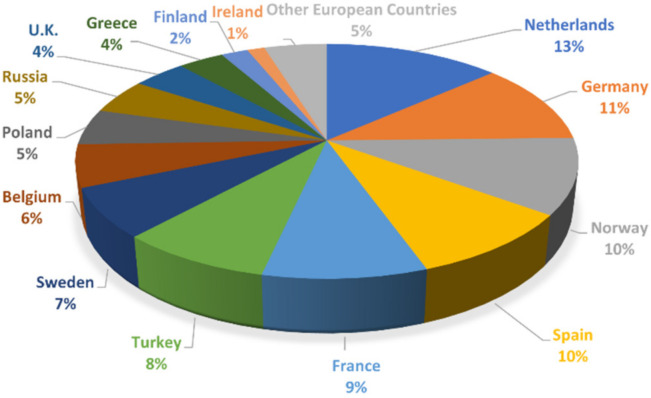
Percentage of new WT installations in 2020 (both on- and offshore), in terms of produced MW capacity. Based on data retrieved from [14].

**Figure 4 sensors-22-01627-f004:**
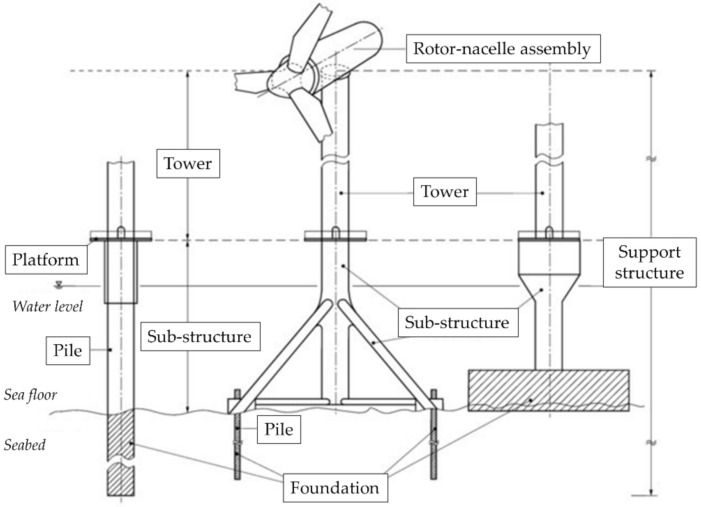
Structural components of a fixed offshore HAWT according to IEC 61400-3-1.

**Figure 5 sensors-22-01627-f005:**
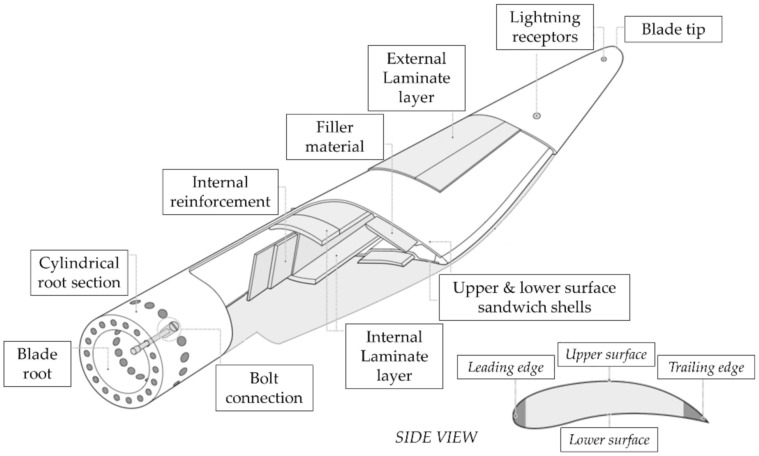
Key components of a typical wind turbine blade. The upper and lower surfaces are also known as the suction (or windward) and pressure (or lee) sides, respectively. The blade root bolt connection shown here is a classic T-bolt type.

**Figure 6 sensors-22-01627-f006:**
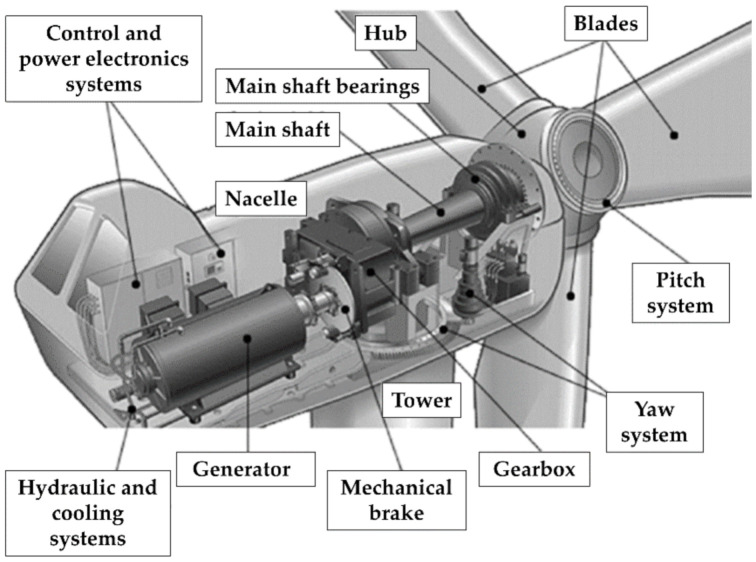
Mechanical and electrical components inside the nacelle of a conventional HAWT.

**Figure 7 sensors-22-01627-f007:**
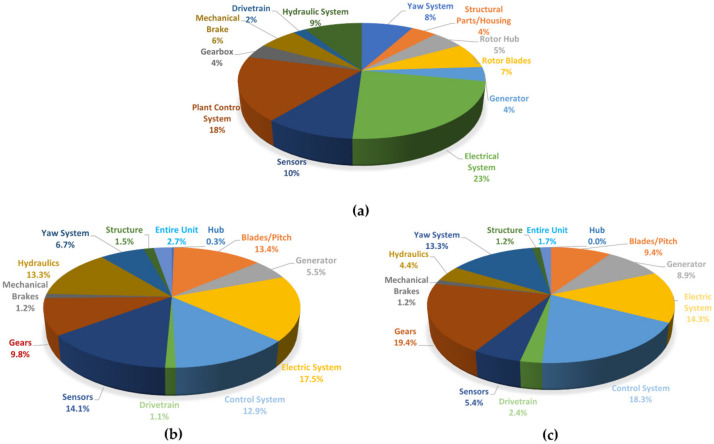
Damage and failure statistics. (**a**) Percentage of unforeseen malfunctions as recorded in Germany for 1500 wind turbines. Based on data retrieved from Ref. [42], collected over 15 years (34,582 events). (**b**,**c**) percentage distribution of the total number of failures and downtime for WTs. Based on data retrieved from Ref. [43], collected from several sources in Sweden, totalling about 600 WTs from 2000 to 2004 (1202 events, 156,202 h). The nomenclature used in the original sources is reproduced for all charts.

**Figure 8 sensors-22-01627-f008:**
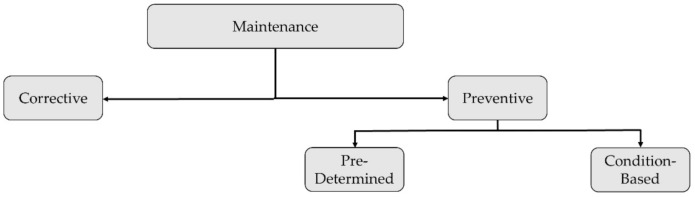
Maintenance strategies according to EN 13306:2017.

**Figure 9 sensors-22-01627-f009:**
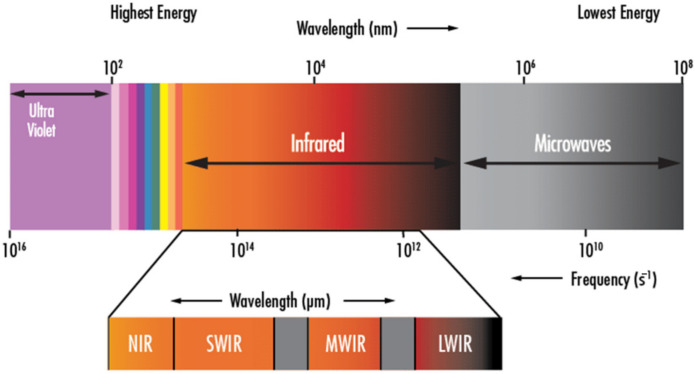
The range of the electromagnetic spectrum that can be covered by common optical techniques (digital, multi/hyperspectral, and thermographic cameras), as well as Gamma-ray, X-ray, microwave, and terahertz testing technologies.

**Figure 10 sensors-22-01627-f010:**
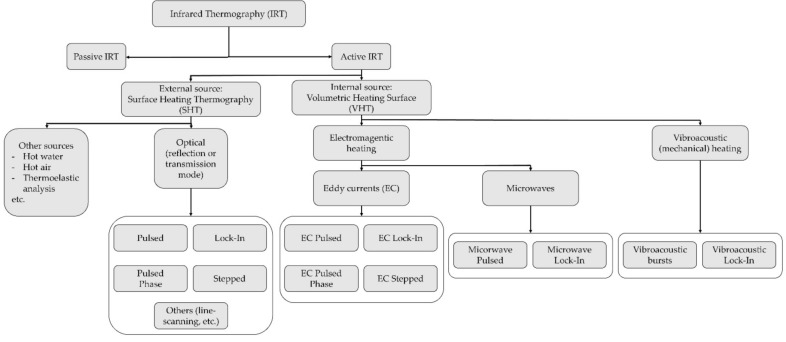
The main IRT techniques available as of 2021.

**Figure 11 sensors-22-01627-f011:**
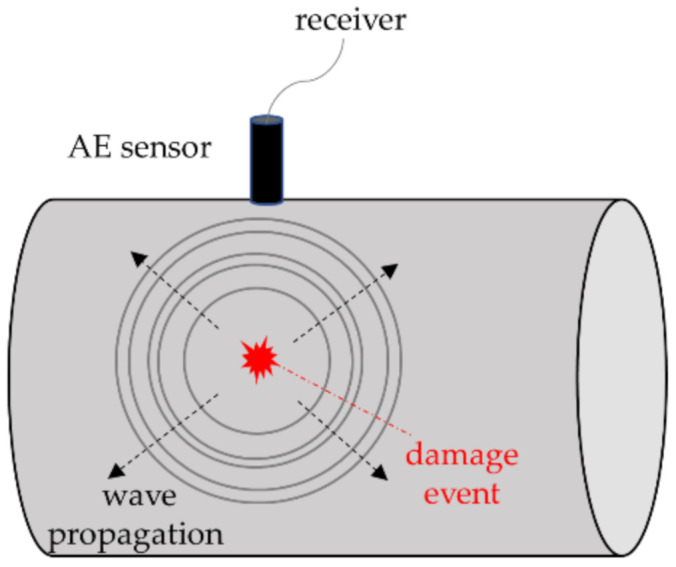
The basic concept of AE event detection.

**Figure 12 sensors-22-01627-f012:**
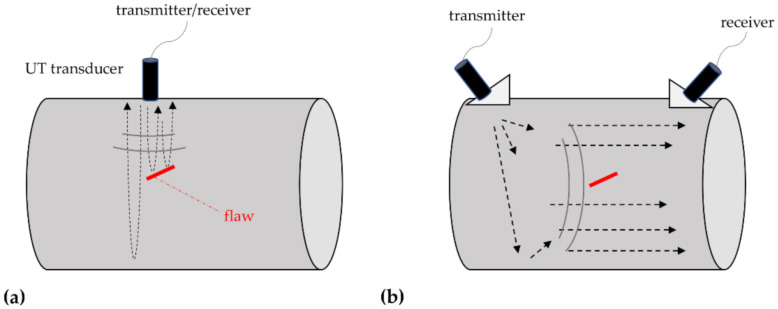
The basic concept of UT. (**a**) conventional, i.e., through the thickness (reflection mode), (**b**) guided waves. (through transmission mode).

**Figure 13 sensors-22-01627-f013:**
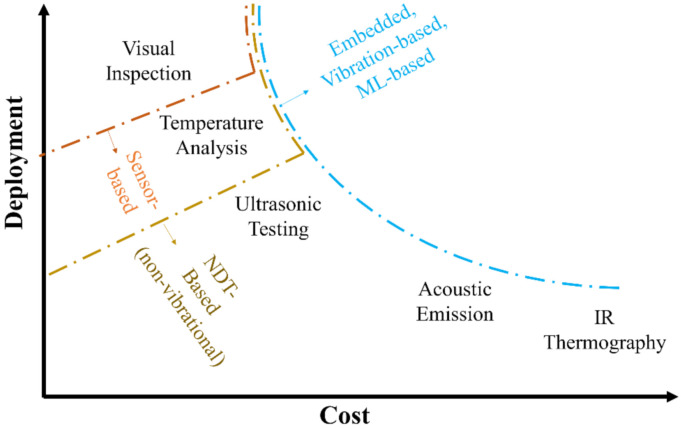
Qualitative distribution of costs and deployment levels of different NDTs. Based on data from Refs. [329,330,331].

**Table 1 sensors-22-01627-t001:** Typical damage typologies in WT blades, according to Refs. [48,49].

Damage Type	Description
#1	Damage formation and growth in the adhesive layer joining the skin and main spar flanges (skin/adhesive debonding and/or the main spar/adhesive layer debonding).
#2	Damage formation and growth in the adhesive layer joining the up- and downwind skins along leading and/or trailing edges (adhesive joint failure between skins).
#3	Damage formation and growth at the interface between the face and core in the sandwich panels in skins and the main spar web (sandwich panel face/core debonding).
#4	Internal damage formation and growth in laminates in the skin and/or main spar flanges, under a tensile or compression load (delamination driven by a tensional or a buckling load).
#5	Splitting and fracture of separate fibres in the laminates of the skin and main spar (fibre failure in tension; laminate failure in compression).
#6	Buckling of the skin due to damage formation and growth in the bond between the skin and main spar under a compressive load. *
#7	Formation and growth of cracks in the gel coat; debonding of the gel-coat from the skin (gelcoat cracking and gel-coat/skin debonding).

* Type #6 can be seen as a particular case of Type #1 damage [50].

**Table 2 sensors-22-01627-t002:** Typical damage typologies in bearings, according to Ref. [55].

Damage Type	Description	Possible causes
Flaking	Creation of regions with a rough and coarse texture due to the splitting off of small pieces from the raceway surface.	Rolling fatigue, caused in turn by excessive load, misalignment, poor lubrification, water or debris inclusions, unsuitable bearing clearance, unevenness in housing rigidity, rust, corrosion pits, dents.
Peeling	Light wear and dull spots on the surface, with micrometric cracks and minor flaking.	Poor or unsuitable lubricant, debris intrusion in the lubricant.
Scoring	Straight lines on the surface, circumferentially on the raceway surface.	Generated by accumulated small seizures, caused in turn by sliding under improper lubrication or excessive/improper loads and conditions (shaft bending, the inclination of inner and outer rings, etc).
Smearing	Surface damage, with the formation of rough and partially melted material.	Generated by accumulated small seizures between bearing components, caused in turn by oil film rupture (because of poor/improper lubrication or high speeds with very light loads).
Fracture	Small pieces broke off due to shock loads or stress accumulation.	Impacts during mounting/dismounting, excessive loads, progression of surface cracks.
Cracks	Formation of surface cracks on the raceway rings and/or rolling elements.	Excessive loads, progression of flaking damage, creep-induced heating, inappropriate shaft (e.g., poor taper angle).
Cage damage	Cage deformation, fracture, and/or wear (considering the cage guide surface, pocket surface, and cage pillars).	Excessive speed, sudden acceleration/deceleration, high temperature, poor lubrication, excessive vibrations, bearing misalignment.
Denting	Small dents on the surface of raceway rings or rolling elements.	Caused by metallic particles or other very small debris caught in the surface during rolling.
Pitting	Pitted surface on the raceway rings or rolling elements.	Poor lubricant, debris in the lubricant, or exposure to moisture.
Wear	Surface deterioration on the raceway rings, rolling elements, cage pockets, and/or roller end faces.	Sliding friction between two surfaces, caused in turn by an irregular motion of the rolling elements, poor lubrication, debris intrusions in the lubricant, or as a progression from chemical or electrical corrosion.
Fretting	Corrosion happening at the contact area between the raceway ring and the rolling elements. It may happen at regular roller pitch intervals.	Repeated sliding on the fitting surface.
False brinelling	Hollow spots that resemble Brinell dents.	Caused by wear, induced in turn by vibration and swaying at the contact points between the raceway and the rolling elements, especially with poor lubrication.
Creep	Shiny appearance on the fitting surface, potentially coupled with scoring and wear.	Relative slipping at the fitting surfaces, due to a loose fit or insufficient sleeve tightening.
Seizure	Softened, deformed, and/or melt material in the raceway rings, rolling elements, or cage.	Excessive load, speed, shaft bending, poor housing or lubrication, small internal clearance.
Electrical corrosion	Corrugations resulting from locally melted material.	Melting by arcing, induced by the passage of electric currents. In turn, these are induced by the electrical potential between the inner and the outer rings.
Pit corrosion	Pits on the surface of raceway rings or rolling elements due to chemical corrosion.	Entry of corrosive gas or liquid, improper lubricant, moisture, high humidity, improper handling and storage conditions.
Mounting flaws	Scratches on the surface of raceway rings or rolling elements caused by mounting/dismounting.	Incorrect mounting/dismounting (impulse loads, the inclination of inner or outer rings, etc).
Discolouration	Discolouration of the cage, rolling elements, or raceway rings.	Poor lubrication and/or high temperature.

**Table 3 sensors-22-01627-t003:** Typical failure modes in wind turbine mechanical components, according to Ref. [61].

Mechanical Component	Common Failure Modes
Gearbox and drive train	Gear tooth damages, high- or low-speed shafts faults, gearbox bearing failures.
Generator	Generator stator failure, generator rotor failure, generator bearing failure.
Main bearing	Bearing failure, bearing rubs, bearing looseness
Pitch gears	Pitch Gear tooth damages.
Yaw gears	Yaw Gear tooth damages.

**Table 4 sensors-22-01627-t004:** Main factors influencing the choice of the maintenance strategy for offshore wind farms.

Study	Year	Mentioned Factors
Henderson et al. [62]	2003	Accessibility of the offshore platform and reliability of the monitoring strategy.
Nielsen et al. [63]	2011	Weather conditions, total power generation, repair strategies, transportation strategies.
Dinwoodie et al. [64]	2012	Repair time, wave height, wind speed, number of wind turbines in the wind farm, ship availability, availability of spare parts stocks.
Scheu et al. [65]	2012	Expected typologies of component failures, ship fleet size, ship type, travel time, number of maintenance workers on staff.
Besnard et al. [66]	2013	Location of accommodation facilities for maintenance staff, vessels for the transfer of crew (type and number), availability of helicopters, organization of work shifts, management of spare parts stocks, technical support, availability of cranes (purchase or contract), environmental conditions (depending on weather and season), economic parameters (electricity prices, ship rental costs).
Halvorsen-Weare et al. [67]	2013	Investment costs, ship costs (fixed and variable costs), failure probability, downtime costs, meteorological data.
Hofmann & Sperstad [68]	2013	Weather conditions (including uncertainty), breakdown rates, electricity price, ship price (costs, fleet composition, type, quantity), workers (shift length, quantity), location of the maintenance base of operations.
Perveen et al. [69]	2014	Protection methodologies, occurrence of cable and component failures, repair strategy, wind speed predictions, and condition monitoring systems.
Endrerud et al. [70]	2015	Weather conditions, ships (availability, operating limits, costs), availability of maintenance technicians, repair times, wind farm layout, cost of spare parts, logistics (warehousing and other costs).
Nguyen & Chou [71]	2018	Duration of maintenance (downtime), expected loss of production during maintenance time, the market price of electricity, location of the wind farm.

**Table 5 sensors-22-01627-t005:** Some notable and recent examples of advanced and automated VI strategies applied for the NDE of wind turbines.

Study	Year	Platform	Computer Vision/Video or Image Processing1 Technique	Application
Stokkeland et al. [91]	2015	Digital camera-equipped multi-copter UAV.	Computer Vision was also utilized for autonomous navigation (moving along the blades to acquire pictures).	SHM
Park et al. [92]	2015	Fixed Digital camera (laboratory test only)	Image segmentation, canny edge detection, and Hough Transform are applied to evaluate the angle changes in the nuts. The method is proposed for bolt loosening monitoring in the ring flange joints in WT towers.	SHM
Wang et al. [93]	2017	Remotely-controlled, digital camera-equipped UAV.	Cascading classifiers (several variants) were applied to detect and locate pixel regions containing cracks in the images.	SHM
Reddy et al. [94]	2019	Digital camera-equipped multi-copter UAV.	A convolutional neural network (CNN) architecture.	SHM
Shihavuddin et al. ^1^ [95].	2019	Digital camera-equipped multi-copter UAV.	Deep learning-based damage detection and classification. Specifically, the authors used the well-established faster region-based CNN (R-CNN) algorithm [96] and compared it to other similar architectures (R-CNN, Fast R-CNN, SSD, and R-FCN).	SHM
Yang et al. [97]	2021	Digital camera-equipped UAV.	The authors used the pre-trained CNNs described in Ref. [98] after integrating them with their image dataset via Transfer Learning ^2^.	SHM

^1^ A dataset of several hundred UAV-taken pictures of a single wind turbine has been released linked to this study [99]. ^2^ Few pictures of wind turbine blades are available easily within common annotated image datasets such as ImageNet and AlexNet.

**Table 7 sensors-22-01627-t007:** Some notable and recent examples of IRT techniques applied for the NDE of wind turbines.

Study	Year	Technique	Notes	Application
Rumsey & Musial [193]	2001	Passive IRT	Infrared thermography was applied by the National Wind Technology Center at the National Renewable Energy Laboratory for the testing of full-size WT blades. One of the tests performed was a fatigue test in which a cyclic load was applied to the WT blade until failure.	SHM
Dattoma et al. [194]	2001	Active IRT (external heating and readings during the cooling phase)	The IRT procedure was experimentally tested on a WT blade sandwich panel, taken from the box spar. Glue infiltration, water ingress, and skin–core debonding were tested.	SHM
Hahn et al. [195]	2002	Thermoelastic stress analysis	Used to monitor the stress distribution on a GFRP blade during static and fatigue tests. Strain gauges were applied as well to assess the integrity of the root section.	SHM
Cheng & Tian [196]	2011	Inductive IRT (pulsed eddy current thermography)	The proposed method is based on inductive thermography for the inspection and assessment of CFRP components.	SHM
Pan et al. [197]	2012	Pulsed eddy current	The inductor and the IR camera were placed on opposite sides to detect damage in the heat transmission mode on CFRP specimens intended for WT blades.	SHM
Cheng & Tian [198]	2013	Pulsed eddy current	Detected surface cracks, impact cracks, defects, and delaminations from transient thermal images or videos on CFRP specimens.	SHM
Dattoma & Giancane [199]	2013	Passive IRT during fatigue tests	Compared DIC and IRT results on a GFRP specimen employed for WT blades.	SHM
Galleguillos et al. [200]	2015	Passive IRT from a UAV platform	Performed in situ surveys on rotating WT blades (in-service) with passive IRT from an unmanned rotorcraft.	SHM
Gao et al. [201]	2016	Pulsed eddy current	Developed a multidimensional tensor model based not only on the analysis of a single physical field such as heat conduction (conventional approach) but also on the inclusion of other properties such as electrical conductivity and magnetic permeability as well.	SHM
Paulmbo et al. [202]	2016	Lock-in IRT analysis (heat source: halogen lamps)	The technique was tested for the debonding of GFRP joints and compared to ultrasonic testing.	SHM
Yang et al. [203]	2016	Pulsed eddy current	Combined eddy current pulsed thermography and thermal-wave-radar analysis for the assessment of delamination on CFRP blades.	SHM
Palumbo et al. [204]	2017	Thermoelastic phase analysis	The study focused on the fatigue damage analysis on GFRP specimens, analysing the thermal signal in the frequency domain.	SHM

**Table 9 sensors-22-01627-t009:** Some notable and recent examples of IRT techniques applied for the NDE of wind turbines.

Study	Year	Technique	Notes	Application
Jørgensen et al. [289]	2004	Ultrasonic immersion test	An early example of UT for the detection of damages and manufacturing defects. The skin, glue, laminate, and sandwich layers were all clearly visible from the scans.	SHM
Jasiüniené et al. [290]	2008	Ultrasonic immersion test with moving water container	A particular type of ultrasonic immersion test (contact pulse–echo immersion testing) was used to assess internal defects in a WT blade. The geometry of the defects was recognized from the ultrasound images obtained.	SHM
Raišutis et al. [291]	2008	Air-coupled guided wave ultrasonic test	The authors used an ultrasonic air-coupled technique to transmit guided waves, locating internal defects in a WT blade.	SHM
Jüngert [292]	2008	Guided wave ultrasonic test	It compared acoustic waves (from hammer tests, using local resonance spectroscopy) with ultrasonic guided waves. Acoustic waves were found to be less subject to scattering and damping while travelling through the fibre-reinforced material but less sensitive to small damages (due to their larger wavelength).	SHM
Jüngert & Grosse [293]	2009	Contact pulse-echo tests	Compared local resonance spectroscopy (from hammer tests) with contact pulse–echo UT on sandwich composites and pristine and delaminated GFRP. Ultrasonic waves correctly detected debonding at adhesive areas.	SHM
Jasiüniené et al. [294]	2009	Air-coupled ultrasonic tests, ultrasonic immersion tests with moving water container, and contact pulse–echo tests	UT and radiographic techniques were compared on WT blade specimens. The ultrasonic techniques proved to be more efficient in terms of implementation as they only require access from one side. The best imaging results, however, were obtained by combining RT and UT techniques.	SHM
Lee et al. [295]	2011	Long distance laser ultrasonic test	To overcome the attenuation due to air travelling, a portable laser-based device was proposed for long-distance UT, up to 40 m (indoor laboratory conditions).	SHM
Park et al. [296]	2013	Long distance laser ultrasonic test	It proposed a new laser ultrasonic imaging technique, specifically intended for rotating blades	SHM
Ye et al. [297]	2014	Pulse-echo test	A portable device for 2D (surface) and 3D (volume) UT scanning was proposed and tested on GFRP WT blade specimens.	SHM
Park et al. [298]	2014	Long-distance laser ultrasonic test	Delamination and debonding were successfully visualized in a GFRP composite wind blade structure.	SHM
Park et al. [299]	2015	Laser ultrasonic propagation imaging system	A two-step UT imaging strategy was proposed, with an initial coarse scanning followed by a second refined one limited to the areas deemed of major interest after the first step. Tested on a 10 kW GFRP WT blade.	SHM
García Marquez & Gómez Muñoz [300]	2020	Macro fibre composite transducers and sinusoidal shaped signals	Cross-correlation and wavelet analysis were applied to detect, assess, and localize delaminations in WT blades.	SHM

**Table 11 sensors-22-01627-t011:** Some notable and recent examples of strain measurements applications for wind turbines.

Study	Year	Notes	Application
Papadopoulos et al. [315]	2000	An early study on the feasibility of static strain measurements for WT blades. The main potential causes of error were discussed and their impact was experimentally estimated.	SHM
Kim et al. [316]	2011	FBG sensors were embedded into a 1/23 scale of the 750 kW composite blade to evaluate its deflection.	SHM
Dimopoulos et al. [317]	2012	The authors used strain measurements from strain gauges to experimentally investigate the buckling behaviour of the thin steel cylindrical shells which make up the HAWT tower.	SHM
Choi et al. [318]	2012	FBG sensors were applied to estimate the static tip deflection of a 100 kW GFRP blade. This shape sensing is intended to avoid potential collisions with the nearby tower.	SHM
Kim et al. [319]	2013	Similar to Choi et al. [318], the authors suggested installing FBG sensors at the bonding line between the shear web and spar cap	SHM
Sierra-Pérez et al. [320]	2016	Compared strain measurements taken from strain gauges, FBG sensors, and Optical Backscatter Reflectometer (OBR) sensors on a prototype GFRP WT blade.	SHM

**Table 12 sensors-22-01627-t012:** Most common monitoring strategies for the different load-bearing and rotating components of a wind turbine, according to the literature review (in particular [61,326,327]) and considering both on- and off-site (laboratory) inspection.

	SHM			CM			
	Tower	Foundations	Blades	Bearings	Shaft	Generator	Gearbox
VI	**X**		**X**		**X** (limited visibility)	**X** (limited visibility)	**X** (limited visibility)
Optical measurements	**X**		**X**				
Shearography			**X**				
IRT	**X**		**X**	**X**	**X**	**X**	**X**
Temperature, non IRT				**X**	**X**	**X**	**X**
X-ray CT	**X**		**X**	**X**			**X**
ET	**X**		**X** (CFRP only)				**X**
AEs	**X**		**X**	**X**	**X**		**X**
UT	**X**		**X**		**X**		
Oil Monitoring				**X**		**X**	**X**
Static strain	**X**	**X**	**X**				

**Table 13 sensors-22-01627-t013:** Advantages and disadvantages of each NDT strategy.

Method	Advantages	Disadvantages
VI	Non-contact Very simple Low cost Does not require extensive training or specific instruments (man-made VI) Can be automated (Computer Vision and autonomous unmanned platforms)	Limited to surface damages and defects. Safety hazard for the personnel (if man-made). Low accuracy and highly subjective (if man-made).
Optical Measurements and Shearography	Non-contact Full-field Relatively fast to perform High sensitivity to damage	Shearography requires a specific (and expensive) setup. Difficult to quantify the extension of damage. Some techniques (e.g., DIC) require surface treatment.
IRT	Non-contact (except vibrothermography) Full-field Relatively fast to perform (except lock-in thermography; depends on the thickness of the material for pulsed and EC pulsed thermography) High sensitivity to damage Many options (surface and volumetric heating, different inputs, etc.) Relatively simple setup (except microwave thermography) Highly standardized (e.g., ISO 10880:2017) Good spatial resolution (depends on the specific option)	Active IRT requires an active source of heating. Only microwaves ensure uniform volumetric heating. Only microwaves and vibrothermography allow selective heating. Only lock-in and pulsed phase thermography are emissivity independent. Surface heating thermography is limited to the outermost layers of the material. Eddy currents cannot be applied to all materials (depending on their conductivity). Pulsed phase thermography requires extensive signal processing to analyze the results. Damage-unrelated factors may cause a rise in temperature. Cannot provide a very accurate damage diagnosis.
Temperature, non IRT	Highly standardized (e.g., ISO 15312:2018).	Requires an embedded sensor (subject to sensor faults). Damage-unrelated factors may cause temperature rise.
X-ray CT	Non-contact Very high spatial resolution	Radiation hazard Complex (and expensive) setup
ET	Non-contact Relatively low-cost.	Sensitive to lift-off Limited by the material conductivity Requires specific instruments.
AEs	Passive (no input required) Able to detect early-stage cracks and small defects. Can be applied on-site and in-service Can be applied also to low-speed rotating machinery. Can cover relatively large areas/volume. High signal-to-noise ratio. Frequency range far from load perturbation.	Relatively expensive. Requires a very high sampling rate. Acoustic wave attenuation in the material. Only detect damages at their inception or during their growth. Difficult to quantify the extension of damage In general, very noisy and difficult to interpret.
UT	Can be applied on-site and in-service Many options Can cover relatively large areas/volume, also with complex geometries	Require an active source of ultrasounds Coupling issues (especially for water-incompatible materials) Ultrasound attenuation in the material The analysis of the results requires an expert user
Oil Monitoring	Easy to install. Enables the direct characterisation of several oil parameters. The results are easy to interpret	Only viable for mechanical systems with a closed-loop oil supply system. Expensive for continuous online monitoring.
Static strain	Can provide both damage detection and shape sensing capabilities. Conventional strain gauges are easy to install. Can be used to monitor dynamic strain as well (for vibration-based inspection).	Fibre optics are still expensive and difficult to install.

## Data Availability

All data reported in this review are available at the referred original sources.

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
