# Peer review of "Non-Destructive Techniques for the Condition and Structural Health Monitoring of Wind Turbines: A Literature Review of the Last 20 Years"

_sensors, 2022, doi:10.3390/s22041627_

Round 1
Reviewer 1 Report
The authors reported the broad literature review of non-destructive evaluation techniques for the condition and structural health monitoring of wind turbines. The contents are detailed and concrete. The paper is in general well-written. However, the following minor concerns should be addressed before it can be accepted for publication:
- In Figure 5, is it a torque wrench for bolt connection or something else? The authors need to explain it.
- In the Section 5.10, the authors should provide the readers with more details about sound directional and damping effects in composites. For example, the following reference talks about the nonlinear resonance technique for fatigue crack detection: Shen, Y.; Cesnik, C.E.S. (2016) “Hybrid local FEM/global LISA modeling of damped guided wave propagation in complex composite structures”, Smart Materials and Structures, 25(9), 095021. In addition, the local ultrasonic resonance spectroscopy approach should be further discussed. For example, the following reference talks about the nonlinear resonance spectral correlation technique for fatigue crack detection: Wang, J.; Shen, Y.; Rao, D.; Xu, W. (2021) “An instantaneous-baseline multi-indicial nonlinear ultrasonic resonance spectral correlation technique for fatigue crack detection and quantification”, Nonlinear Dynamics, 103(1), 677-698. The authors are suggested to add these references in this section for the readers’ benefit.
- The first letter of the sentence should be capitalized, e.g., the last two lines of table 3, and the first sentence of the third paragraph in Section 4.2. The authors need to revise this aspect.
-
J. Yang, J. He, X. Guan, D. Wang, H. Chen, W. Zhang, Y. Liu, A probabilistic crack size quantification method using in-situ Lamb wave test and Bayesian updating, Mechanical Systems and Signal Processing, 78 (2016) 118-133.
-
A. Eremin, E. Glushkov, N. Glushkova, R. Lammering, Guided wave time-reversal imaging of macroscopic localized inhomogeneities in anisotropic composites, Structural Health Monitoring, (2019).
Otherwise, the reviewer finds the paper quite pleasant to read. It will be a good contribution to the SHM community.
Author Response
REVIEWER #1.
The authors reported the broad literature review of non-destructive evaluation techniques for the condition and structural health monitoring of wind turbines. The contents are detailed and concrete. The paper is in general well-written.
However, the following minor concerns should be addressed before it can be accepted for publication:
- In Figure 5, is it a torque wrench for bolt connection or something else? The authors need to explain it.
Reply: We added a more detailed description in the caption of Figure 5, many thanks for your suggestion.
- In the Section 5.10, the authors should provide the readers with more details about sound directional and damping effects in composites. For example, the following reference talks about the nonlinear resonance technique for fatigue crack detection: Shen, Y.; Cesnik, C.E.S. (2016) “Hybrid local FEM/global LISA modeling of damped guided wave propagation in complex composite structures”, Smart Materials and Structures, 25(9), 095021. In addition, the local ultrasonic resonance spectroscopy approach should be further discussed. For example, the following reference talks about the nonlinear resonance spectral correlation technique for fatigue crack detection: Wang, J.; Shen, Y.; Rao, D.; Xu, W. (2021) “An instantaneous-baseline multi-indicial nonlinear ultrasonic resonance spectral correlation technique for fatigue crack detection and quantification”, Nonlinear Dynamics, 103(1), 677-698. The authors are suggested to add these references in this section for the readers’ benefit.
Reply: After carefully studying the mentioned papers, we agree that these can be a useful addition to our peer review. Thus, we added them to our Section 5.10. Many thanks for your suggestion.
- The first letter of the sentence should be capitalized, e.g., the last two lines of table 3, and the first sentence of the third paragraph in Section 4.2. The authors need to revise this aspect.
Reply: We double-checked throughout the whole text for these and other typos and grammatical mistakes. Many thanks for pointing that out.
- J. Yang, J. He, X. Guan, D. Wang, H. Chen, W. Zhang, Y. Liu, A probabilistic crack size quantification method using in-situ Lamb wave test and Bayesian updating, Mechanical Systems and Signal Processing, 78 (2016) 118-133.
- A. Eremin, E. Glushkov, N. Glushkova, R. Lammering, Guided wave time-reversal imaging of macroscopic localized inhomogeneities in anisotropic composites, Structural Health Monitoring, (2019).
Reply: After carefully studying the two papers mentioned in Remarks #4 and #5, we agree that these can be a useful addition to our peer review. Thus, we added them to our Section 5.10, where guided wave ultrasonic testing is discussed. Many thanks for your suggestions.
Otherwise, the reviewer finds the paper quite pleasant to read. It will be a good contribution to the SHM community.
Final comments to the reviewer: We would like to thank Reviewer #1 for his/her support and constructive comments.
Reviewer 2 Report
The manuscript presented a survey of non-destructive techniques (NDT) for condition and structural health monitoring of wind turbines in the recent 20 years. It includes four topics of NDT for wind turbines, which are the politics and economics, structural and mechanical components, applications for non-destructive evaluation techniques, and major non-destructive evaluation techniques. The review is comprehensive and generally well written. Some comments/suggestions are listed below for improving the quality of the work.
- In Figure 1, it is suggested to provide worldwide global figures, including those countries with high GDP.
- The complete forms of some abbreviations are missing at their first appearance. For example, GW appeared in Section 2.1 but “gigawatts (GW)” was appeared in Section 2.2.
- The symbol “÷” on line 256, 257, 259, 260, 413, and 534 should be “-” that refer “to”.
- There should be no full stop punctuation at the end of the title of the sections and subsections.
- A number of references do not have precious information. Please modify.
- In line 517, the first letter should be capital.
Author Response
REVIEWER #2.
The manuscript presented a survey of non-destructive techniques (NDT) for condition and structural health monitoring of wind turbines in the recent 20 years. It includes four topics of NDT for wind turbines, which are the politics and economics, structural and mechanical components, applications for non-destructive evaluation techniques, and major non-destructive evaluation techniques. The review is comprehensive and generally well written. Some comments/suggestions are listed below for improving the quality of the work.
In Figure 1, it is suggested to provide worldwide global figures, including those countries with high GDP.
Reply: Unfortunately, we based our Figure 1 on data retrieved from the report “Wind energy in Europe - 2020 Statistics and the outlook for 2021-2025”, which accounts exclusively for countries included in the European continent. We better remarked in the text that those paragraphs are only limited to the near-future European market. We added some more information, retrieved from other sources, regarding the worldwide developed (high GDP) countries. Thank you for your suggestion.
The complete forms of some abbreviations are missing at their first appearance. For example, GW appeared in Section 2.1 but “gigawatts (GW)” was appeared in Section 2.2.
Reply: We double-checked throughout the whole text for these and other typos and grammatical mistakes. Many thanks for pointing that out.
The symbol “÷” on line 256, 257, 259, 260, 413, and 534 should be “-” that refer “to”.
Reply: We modified our text accordingly, thank you for pointing that out.
There should be no full stop punctuation at the end of the title of the sections and subsections.
Reply: We corrected them accordingly, thank you for pointing that out.
A number of references do not have precious information. Please modify.
Reply: We added the missing information, thank you for pointing that out.
In line 517, the first letter should be capital.
Reply: We modified our text accordingly, thank you for pointing that out.
Final comments to the reviewer: We would like to thank Reviewer #1 for his/her support and constructive comments
Reviewer 3 Report
The authors aims to review the recent progresses on SHM/NDE of wind turbines. The topic is important and a comprehensive review on such topic is necessary since the rapid growth of research works on it. Overall the paper is organized reasonably and gives the reader a macro understanding of related fields. Two questions are presented here for authors consideration.
Q1: In Section 2, the related economical costs and return for holders to apply SHM/NDE techniques to theirs wind turbines should be discussed.
Q2: In Sections 3-5, bolt loosening failure in tower due to dynamic load occurs often. The discussion on such failure mode and related SHM/NDE methods are not sufficient.
Author Response
REVIEWER #3.
The authors aims to review the recent progresses on SHM/NDE of wind turbines. The topic is important and a comprehensive review on such topic is necessary since the rapid growth of research works on it. Overall the paper is organized reasonably and gives the reader a macro understanding of related fields. Two questions are presented here for authors consideration.
Q1: In Section 2, the related economical costs and return for holders to apply SHM/NDE techniques to theirs wind turbines should be discussed.
Reply: We added a discussion about this important aspect in our Section 2. Many thanks for your suggestion.
Q2: In Sections 3-5, bolt loosening failure in tower due to dynamic load occurs often. The discussion on such failure mode and related SHM/NDE methods are not sufficient.
Reply: We included a more detailed discussion about this specific failure mode in our Section 3.3 (Incidence and main causes of structural collapse). Many thanks for your suggestion.
Final comments to the reviewer: We would like to thank Reviewer #1 for his/her support and constructive comments.